

# Variations in $O_3$, CO, and $CH_4$ over the Bay of Bengal during the summer monsoon season: Ship-borne measurements and model simulations

Imran A. Girach[1,2], Narendra Ojha[2], Prabha R. Nair[1], Andrea Pozzer[2], Yogesh K. Tiwari[3], K. Ravi Kumar[4,5], and Jos Lelieveld[2]

[1]Space Physics Laboratory, Vikram Sarabhai Space Centre, Thiruvananthapuram 695022, India

[2]Department of Atmospheric Chemistry, Max Planck Institute for Chemistry, Mainz 55128, Germany

[3]Indian Institute of Tropical Meteorology, Pune 411 008, India

[4]National Institute of Polar Research, Tachikawa, Japan

[5]Department of Environmental Geochemical Cycle Research, JAMSTEC, Yokohama, Japan

*Correspondence to*: Imran A. Girach (imran.girach@gmail.com) and Narendra Ojha (narendra.ojha@mpic.de)

## Abstract

We present ship-borne measurements of surface ozone, carbon monoxide and methane over the Bay of Bengal (BoB), the first time such measurements have been taken during the summer monsoon season, as a part of the Continental Tropical Convergence Zone (CTCZ) experiment during 2009. $O_3$, CO, and $CH_4$ mixing ratios exhibited significant spatial and temporal variability in the ranges of 8–54 nmol mol$^{-1}$, 50–200 nmol mol$^{-1}$, and 1.57–2.15 µmol mol$^{-1}$, with means of 29.7±6.8 nmol mol$^{-1}$, 96±25 nmol mol$^{-1}$, and 1.83±0.14 µmol mol$^{-1}$, respectively. The average mixing ratios of trace gases over northern BoB ($O_3$: 30±7 nmol mol$^{-1}$, CO: 95±25 nmol mol$^{-1}$, $CH_4$: 1.86±0.12 µmol mol$^{-1}$), in airmasses from northern or central India, did not differ much from those over central BoB ($O_3$: 27±5 nmol mol$^{-1}$, CO: 101±27 nmol mol$^{-1}$, $CH_4$: 1.72±0.14 µmol mol$^{-1}$), in airmasses from southern India. Spatial variability is observed to be most significant for $CH_4$. The ship-based observations, in conjunction with backward air trajectories and ground-based measurements over the Indian region, are analyzed to estimate a net ozone production of 1.5–4 nmol mol$^{-1}$ day$^{-1}$ in the outflow. Ozone mixing ratios over the BoB showed large reductions (by ~20 nmol mol$^{-1}$) during four rainfall events. Temporal changes in the meteorological parameters, in conjunction with ozone vertical profiles, indicate that these low ozone events are associated with downdrafts of free-tropospheric ozone-poor airmasses. While the observed variations in $O_3$ and CO are successfully reproduced using the Weather Research and Forecasting model with Chemistry (WRF-Chem), this model overestimates mean concentrations by about 20%, generally overestimating $O_3$ mixing ratios during the rainfall events. Analysis of the chemical tendencies from model simulations for a low-$O_3$ event on August 10, 2009, captured successfully by the model, shows the key role of horizontal advection in rapidly transporting ozone-rich airmasses across the BoB. Our


study fills a gap in the availability of trace gas measurements over the BoB, and when combined with data from
previous campaigns, reveals large seasonal amplitude (~39 and ~207 nmol mol$^{-1}$ for $O_3$ and CO, respectively) over
the northern BoB.

## 1.  Introduction

Tropospheric ozone ($O_3$) is the third most important greenhouse gas, contributing to global warming and climate
change with its radiative forcing of $0.40\pm0.20$ Wm$^{-2}$ (IPCC 2013). $O_3$ is also a pivotal trace gas in tropospheric
chemistry, as it is a major source of hydroxyl radical (OH), which removes most of the organic compounds and
pollutants from the atmosphere and controls the oxidation capacity of the troposphere (e.g. Brasseur et al., 1999;
Finlayson-Pitts and Pitts, 2000; Seinfeld and Pandis, 2006). Further, enhanced concentrations of surface $O_3$ have
detrimental effects on human health and vegetation (Heagle, 1989; Seinfeld and Pandis, 2006). Approximately 80%
of tropospheric $O_3$ is produced by in situ photochemical reactions in the presence of nitrogen oxides ($NO_x = NO +
NO_2$) involving the precursor gases of methane, non-methane hydrocarbons (NMHCs), and CO (Fishman et al.,
1979; Crutzen et al., 1999; Seinfeld and Pandis, 2006). The remaining 20% of tropospheric ozone is attributed to
intrusions of stratospheric air during frontal activities or to tropopause folding events (Lelieveld and Dentener, 2000;
Sprenger et al., 2007). Depending upon meteorological conditions and the availability of the aforementioned
precursors, a net production or destruction of $O_3$ prevails. The average lifetime of ozone is about one week in the
lower troposphere, which leads to large variability in its spatial and temporal distributions, as compared to the long-
lived greenhouse gases. The budget of tropospheric ozone and its implications for human health, crop yields, and
climate are, however, not yet well quantified, especially over regions in Asia. This is mainly due to insufficient in
situ measurements (e.g. Cooper et al., 2014; Monks et al., 2015).

Carbon monoxide (CO) is an indirect greenhouse gas which also has adverse effects on humans and animals (WHO
1999). Although it does not have a direct greenhouse effect like methane or carbon dioxide, its role in atmospheric
chemistry is estimated to cause an indirect radiative forcing of 0.23 (0.18–0.29) Wm$^{-2}$ (IPCC 2013). The major
sources of CO are fossil fuel combustion, biomass burning, and oxidation of hydrocarbons such as $CH_4$ and isoprene
(e.g. Jacob, 1999; Bergamaschi et al., 2000; Seinfeld and Pandis, 2006).

Methane ($CH_4$) is one of the major greenhouse gases, with a direct radiative forcing of $0.48\pm0.05$ Wm$^{-2}$ (IPCC
2013). This gas plays a major role in the climate and in atmospheric chemistry. $CH_4$ is emitted from variety of
natural and anthropogenic sources (Jacob, 1999) and is removed primarily through its reaction with OH radicals
(Fung et al., 1991, Seinfeld and Pandis, 2006).

The marine regions adjoining South Asia, despite being far from direct anthropogenic activities, have been observed
to have elevated levels of surface $O_3$ due to the outflow of continental pollution (Lawrence and Lelieveld, 2010) and
minimal chemical loss by titration (e.g. Lal and Lawrence, 2001; Ojha et al., 2012). Suggested sources for this
elevated ozone and other trace gases observed over the marine regions surrounding India are anthropogenic, biomass
burning, and biogenic emissions over continental India (Naja et al., 2004; Lawrence and Lelieveld, 2010; Nair et al.,





2011; David et al., 2011). The airmasses exposed to continental emissions undergo chemical transformation, including ozone production, during their transport to the cleaner marine regions. In situ measurements over the marine regions adjacent to the South Asia are therefore required to understand the chemistry in the airmasses transported to the marine regions, the effects of direct outflow, and en route chemical transformation. (Lawrence and Lelieveld, 2010, and references therein).

The experiments that have been conducted to date over the marine environment adjacent to the Indian region have revealed considerable spatial heterogeneity in the distribution of trace gases and aerosols, influences from source regions such as the Indo-Gangetic Plains (IGP), and radiative impacts (Nair et al., 2011; David et al., 2011; Mallik et al., 2013; Moorthy et al., 2009; Nair et al., 2010). Observations made during the Indian Ocean Experiment (INDOEX; Lal and Lawrence, 2001) and model simulations (Ojha et al., 2012) both found the ozone mixing ratios over these remote marine regions to be even higher than those over the upwind continental regions, due to complex ozone chemistry. Lawrence and Lelieveld (2010) provided a detailed review of the outflow of trace gases and aerosols from South Asia to the surrounding marine regions. Both the export of continental airmasses and the transport of marine air to the continental regions have strong seasonal dependence associated with the changes in synoptic scale dynamics and monsoonal circulations (e.g. Kumar et al., 2015).

The marine environment of the Bay of Bengal (BoB), the largest bay in the world, is surrounded by landmasses on three sides, making it highly conducive for the accumulation of trace species. Further, seasonal changes in synoptic winds make this a unique region to study variations in trace species due to transport and en route photochemistry. Considering the aforementioned special characterisitics of the BoB, as well as the considerable heterogeneity of trace gas and aerosol distribution, in situ measurements covering large areas are essential for investigating the distribution of pollutants and the controlling processes. Extensive in situ measurements of various trace gases over the BoB have been conducted in the following field campaigns: INDOEX during the winter months of 1998 and 1999 (Lelieveld et al., 2001; Muhle et al., 2002); the Integrated Campaign for Aerosols, gases, and Radiation Budget (ICARB) during the March–May (pre-monsoon season) of 2006 (Nair et al., 2011; Srivastava et al., 2011; Srivastava et al., 2012); the winter-ICARB (W-ICARB) during December–January 2009 (Girach and Nair, 2010; 2014; David et al., 2011); the Bay of Bengal Experiment (BOBEX)-I during February–March 2001 (Lal et al., 2006); the Bay of Bengal Process Studies (BOBPS) during September–October 2002 (Sahu et al., 2006); BOBEX–II during February 2003 (Lal et al., 2007); and an unnamed campaign conducted during October–November 2010 (Mallik et al., 2013).

Although earlier studies have covered the spatio-temporal distribution of trace gases during most seasons over the BoB, there is still a lack of observations over the BoB during the summer monsoon season (June–August). Asian summer monsoon circulation provides a pathway for pollution transport into the stratosphere (Randel et al., 2010), and observations taken during monsoon season capture a time of high water-vapour loading over the BoB, which influences the oxidation capacity of the atmosphere. Deep convection during the summer monsoon can uplift boundary layer pollution to higher altitudes, which is then distributed over a larger region, thereby influencing air quality and climate over much larger regions (Lawrence and Lelieveld, 2010), extending as far as, for example, over the Mediterranean (e.g. Lelieveld et al., 2002; Scheeren et al., 2003). Such in situ measurements are also essential



given the fact that remote sensing of boundary layer ozone using, for example, the Tropospheric Emission Spectrometer (www.tes.jpl.nasa.gov) on board the Aura satellite, has relatively higher uncertainty (Verstraeten et al., 2013). The uncertainties in satellite retrievals of trace species are particularly high during the summer monsoon season, as the view of satellite instruments is frequently obscured by thick clouds.

In the present paper, we present ship-based measurements of surface $O_3$, CO, and $CH_4$ over the BoB for the year 2009, the first time such measurements have been taken during the summer monsoon season in this region. These observations were carried out as a part of the Continental Tropical Convergence Zone (CTCZ) experiment (http://odis.incois.gov.in/index.php/project-datasets/ctcz-programme) under the Indian Climate Research Programme (ICRP) of the Government of India. In this study, we analyse the spatial and temporal variations in

ozone over the BoB and the effects of transport. These observations are compared with simulations from a regional model, Weather Research and Forecasting coupled with Chemistry (WRF-Chem). We investigate sharp reductions that we have observed in $O_3$ during rainfall events in greater detail.

The manuscript begins with a description of the ship cruise in Section 2, followed by experimental setup and observations in Section 3, description of model simulations in Section 4, and results of the study in Section 5. A

summary and main conclusions are presented in Section 6.

**2.    The cruise track and background conditions**

Figure 1 shows the cruise track of the Oceanic Research Vessel (RV) *Sagar Kanya* during the CTCZ experiment (cruise number SK 261). The arrows show the direction of the ship, which sailed from Chennai (80.3° E, 13.1° N)

on July 16, 2009. The cruise offered greater coverage in the northern BoB than the southern or central BoB areas. To take time series measurements, the ship was kept stationary at 89° E, 19° N for fifteen days (July 22 to August 06, 2009). After several tracks, covering latitude sector 11.0 to 21.1° N and longitude sector 80.3 to 90.1° E, the cruise ended on  August 17, 2009 at Chennai, for a total of 32 days of voyage. The average prevailing wind patterns at 925 hPa during the cruise period are obtained from NCEP/NCAR reanalysis (http://www.esrl.noaa.gov/psd; Fig. 1). The

prevailing westerly or south-westerly winds are conducive for the transport of ozone and its precursors from the Indian landmass to the BoB during the summer monsoon season. The spatial distribution of emissions of $NO_x$, an ozone precursor gas, is also shown as colour map in Fig. 1. $NO_x$ emissions are obtained from the Intercontinental Chemical Transport Experiment – Phase B (INTEX-B) inventory (Zhang et al., 2009), which is representative of the year 2006. Relatively high $NO_x$ emissions are located over parts of eastern and southern India.


**3.    Experimental details and data**

Surface $O_3$ measurements were carried out using an online ultraviolet (UV) photometric ozone analyzer (Model O3 42), manufactured by Environnement S.A, France. The analyser utilises the absorption of UV radiation by ozone molecules at 253.7 nm and derives ozone mixing ratios using the Beer–Lambert law. This UV absorption-based

analyser has an uncertainty of about 5% (Tanimoto, 2007), corresponding to ~1.5 nmol mol$^{-1}$ for the observed range of ozone. Zero noise of the instrument is 0.5 nmol mol$^{-1}$. The instrument has a lower detection limit of 1 nmol mol$^{-1}$





and a linearity of ±1%. An individual measurement is performed at a minimum response time of 10 seconds. The analyzer was operated on auto-response mode, whereby responses could 10–90 seconds depending upon changes in ozone mixing ratios. However, data were recorded continuously at 5-minute intervals.

CO measurements were made using an online CO analyzer (Model CO12 Module) manufactured by Environment S.A, France. This instrument was based on the principle of Non-Dispersive Infrared (NDIR) absorption by CO molecules at the wavelength of 4.67 μm. The instrument has a lower detection limit of 50 nmol mol$^{-1}$, a linearity of 1%, and a response time of 40 seconds. The overall uncertainty in hourly CO measurements is estimated to be ~ 10% at a CO value of 150 nmol mol$^{-1}$ (Sawa et al., 2007; Tanimoto et al., 2007).

Air was drawn from a height of approximately 15 meter above the sea surface through a Teflon tube. Before and after the cruise, both analyzers were calibrated, with calibration factors not found to be significantly changed. Meteorological parameters such as pressure, temperature, and relative humidity were measured continuously onboard the ship. Trace gas measurements affected by the ship exhaust were identified and discarded using onboard wind direction measurements.

In addition, a total of 29 air samples were collected in 1-liter glass flasks during the cruise and were analyzed for methane using a Gas Chromatograph (GC) coupled with a Flame Ionization Detector (FID), as described in Tiwari and Ravi Kumar (2011). These methane measurements are traceable to the WMO standard scale. Methane standards were obtained from the WMO Central Calibration Laboratory (CCL) at the National Oceanic and Atmospheric Administration (NOAA)/Earth System Research Laboratory (ESRL)/Global Monitoring Division (GMD), located in

Boulder, Colorado, USA. The precision for methane measurements was approximately ±0.1 μmol mol$^{-1}$. A detailed description of the analytical procedure for methane measurement is given in Ravikumar et al. (2014).

Using the same ozone analyzer as the one used for surface O$_3$ measurements over BoB, continuous measurements of surface O$_3$ were taken at Thumba, Thiruvananthapuram (David and Nair, 2011; Girach et al., 2012) in July 2011. Along with various meteorological parameters, rainfall measurements were also taken at Thumba at 5-minute

accumulation intervals using an automatic weather station manufactured by Dynalab Weathertech Pvt. Ltd, India. The site, Thumba, is situated just ~500 m away from the west coast, with sandy terrain, and is a less populated area in the city of Thiruvananthapuram (8.5° N, 76.9° E) at southern tip of India. For more details about the Thumba site and measurements please see, for example, Nair et al. (2002) and David and Nair (2011).

A vertical profile of O$_3$ was measured on July 28, 2011 at Thumba using an electrochemical concentration cell

ozonesonde (EN-SCI 2ZV7 ECC; Komhyr, 1969, 1995). The accuracy of such ozonesondes is reported to be about ±5-10 % up to ~30 km (Smit et al., 2007). More details of this measurement technique can be found in Ojha et al. (2014).

The accumulated rainfall for every 3-hour interval from the Tropical Rainfall Measuring Mission (TRMM; with a horizontal grid size of 0.25° × 0.25°) is also utilized in this study to complement the onboard rainfall measurements.

The 3B42 algorithm is used to calculate precipitation and root-mean-square precipitation-error estimates; these two estimates were then used to compute hourly and daily rainfall estimates (Huffman et al., 1995).

## 4.    Model Simulations



The Weather Research and Forecasting model with Chemistry (WRF-Chem; Grell et al., 2005) version-3.5.1 was
used to simulate meteorological and chemical fields during the campaign period. The model domain (Fig. 2) is
defined on the Mercator projection, centred at 86° E, 16° N, at a spatial resolution of 15 km x 15 km. The model has
51 vertical levels from surface to 10 hPa. The simulations were conducted for the period of June 29 to August 31,
2009, covering the complete measurement period. The meteorological inputs have been adopted from ERA-interim
reanalyses by the ECMWF. Horizontal winds, temperature, and water vapour are nudged above the planetary
boundary layer using a nudging coefficient of 0.0003 s$^{-1}$ (Kumar et al., 2015), employing the Four Dimensional Data
Assimilation (FDDA) technique. Anthropogenic emissions of CO, NO$_x$, SO$_2$, and NMVOCs are provided by a
regional emission inventory that was developed to support the Intercontinental Chemical Transport Experiment –
Phase B (INTEX-B; Zhang et al., 2009; Kumar et al., 2012b). This inventory is representative of the year 2006.
Aerosol emissions are provided by the Hemispheric Transport of Air Pollution (HTAP v2) inventory (Janssens-
Maenhout et al., 2015). Biomass burning emissions from NCAR Fire Inventory (FINN; Wiedinmyer et al., 2011),
and biogenic emissions calculated online using MEGAN (Guenther et al., 2006) were used in the simulations.

Gas-phase chemistry in the model is represented by the second-generation Regional Acid Deposition Model
(RADM2; Stockwell et al., 1990), and the aerosol module is based on MADE SORGAM (Binkowski and Shankar,
1995; Ackermann et al., 1998; Schell et al., 2001). Initial and boundary conditions for chemical fields are provided
by the MOZART-4/GEOS5 data. The options used to parameterize different atmospheric processes are given in
Table 1. For more information about meteorological nudging, chemical mechanisms, emissions, boundary
conditions, and evaluation of WRF-Chem against in situ measurements and satellite data over the Indian region,
please see, for example, Kumar et al. (2012a; 2012b; 2015), Ansari et al. (2016), and Ojha et al. (2016). Model-
simulated mean spatial distributions of O$_3$ and CO over the model domain during the study period are shown in
Figure 2.

## 5. Results and Discussion

### 5.1 Variations in O$_3$, CO, and CH$_4$ over the BoB

Figure 3 shows the observed variations in O$_3$, CO, and CH$_4$ along the ship track during the July 16 to August 17,
2009 period of the summer monsoon season. In Figure 3, the solid black lines define two regions, central BoB (80-
91° E, 11-16° N) and northern BoB (81-91° E, 16-21.5° N). All of the measured trace gases show significant spatio-
temporal heterogeneity over the BoB region during the summer monsoon season. O$_3$ levels are found to vary from as
low as 8 nmol mol$^{-1}$ to as high as 54 nmol mol$^{-1}$, with average mixing ratio derived from the complete data of
29.7±6.8 nmol mol$^{-1}$. CO mixing ratios are observed to be in the range of 50 nmol mol$^{-1}$ falling below the detection
limit of the instrument to 198 nmol mol$^{-1}$, with an average value of 96±25 nmol mol$^{-1}$ from all observations. As CO
mixing ratios below the detection limit of the instrument are discarded from the analysis, the reported minimum and
average values of CO mixing ratios are therefore slightly higher than their actual values. CH$_4$ mixing ratios are
observed to range from 1.57–2.15 µmol mol$^{-1}$, with average of 1.83±0.14 nmol mol$^{-1}$. We further separated the
observations into two defined geographical regions: northern BoB and central BoB (Figure 3). The average mixing
ratios for O$_3$, CO, and CH$_4$ are observed to be ~ 30±7 nmol mol$^{-1}$, 95±25, nmol mol$^{-1}$, and 1.86±0.12 µmol mol$^{-1}$,



respectively, over northern BoB. These ratios are comparable or only slightly higher than those over central BoB: $O_3$: 27±5 nmol mol$^{-1}$, CO: 101±27nmol mol$^{-1}$, and $CH_4$: 1.72±0.14μmol mol$^{-1}$. Average $CH_4$ mixing ratios, however, showed a significant difference of 0.14 μmol mol$^{-1}$ between northern and central BoB during the summer monsoon season.

In addition to sailing across the BoB, the ship was also kept stationary for fifteen days, from July 22 to August 06, 2009 at 89° E, 19° N. During this time period, surface $O_3$, CO, and $CH_4$ mixing ratios are observed to fall into the range of 9–46 nmol mol$^{-1}$, 58–144 nmol mol$^{-1}$, and 1.71–1.89 μmol mol$^{-1}$, respectively, with temporally averaged mixing ratios of 28±7 nmol mol$^{-1}$, 91±19 nmol mol$^{-1}$, and 1.81±0.06 μmol mol$^{-1}$, respectively.

The HYbrid Single Particle Lagrangian Integrated Trajectory (HYSPLIT) model was used to simulate five-day

backward airmass trajectories arriving at 500 m (a height that falls within the marine atmospheric boundary layer) above the measurement locations (Draxler and Rolph, 2003; Rolph, 2003; http://www.arl.noaa.gov/ready.html), as shown in the Fig. 4. Trajectories are colour-coded to show the altitude variations of the air-parcel along its path. The influences of two different airmasses are observed over the BoB during the CTCZ experiment. Over central BoB, the backward air trajectories are seen to cross southern India (i.e. <13° N), where a belt of elevated anthropogenic

emissions (5–20 mol km$^{-2}$ hr$^{-1}$ of $NO_x$; see Fig. 1) is located. In contrast, most of the air trajectories over northern BoB come through the central Indian region, where anthropogenic emissions are significantly lower. For example, with the exception of a few hotspots, $NO_x$ emissions above 13° N are in the range of 1–10 mol km$^{-2}$ hr$^{-1}$ (Fig. 1).

The observed spatio-temporal variations in the trace gases are investigated by calculating the percentage residence time of airmasses over land, using HYSPLIT simulated 5-day backward air trajectories. Figure 5a–c shows the

temporal variations in $O_3$, CO, and $CH_4$ during the CTCZ experiment along the cruise track. The percentage residence time of airmasses over continental India is also shown (blue line), as estimated by the ratio of residence time over land to the total trajectory time of 5 days. Red vertical bars depict the sharp reductions in $O_3$ as well as CO mixing ratios associated with rainfall events (see Section 5.4). Similar variations in $O_3$ mixing ratios and residence time over continental India indicate the influences of transport from the Indian subcontinent on the observed spatio-

temporal variations over the BoB during the summer monsoon season. The occasions on which such a one-to-one correspondence are not observed can be attributed to varying source strengths, vertical mixing or dilution, and en route photochemistry. As seen in Fig. 5b, CO is also associated with residence time, although not as strongly as in the case of $O_3$. $CH_4$ does not show a considerable correlation with residence time over the Indian subcontinent (Fig. 5a).

Generally, during the summer monsoon season, relatively cleaner marine airmasses from the Arabian Sea are transported to the Indian region. These airmasses are then exposed to regional emissions and subjected to photochemistry depending upon the availability of solar insolation under the cloudy conditions of monsoon. The airmasses in which precursors have accumulated, and to some extent photo-chemically processed, outflows into the BoB. As a result, the airmasses out-flowing at the eastern coast of India could have higher ozone mixing ratios than

the background air coming from the Arabian Sea into the western coast of India. The difference between these two values is a representative of the ozone build-up that can be attributed to regional pollution; this difference would also reflect the extent of photochemical processing that would have taken place.



As the observational site Thumba, Thiruvananthapuram, is situated just at the Arabian Sea coast, the monsoon-time observations here could be approximated to represent the background ozone mixing ratios entering from the Arabian Sea. In August 2009, using only daytime monthly average $O_3$, the ozone at Thumba during the monsoon season was observed to be 23±7 nmol mol$^{-1}$. Since the objective of investigation is the additional $O_3$ over the BoB produced by en route photochemistry, daytime $O_3$ values at Thiruvananthapuram are therefore compared with all the observations over the BoB. The average mixing ratio observed over the BoB during monsoon season for July 16–August 17, 2009 was 30±7 nmol mol$^{-1}$, which was ~7 nmol mol$^{-1}$ higher than the Arabian Sea airmass. This additional amount of ~7 nmol mol$^{-1}$ could be attributed to the effects of regional and en route photochemical ozone production. Net ozone production rate in the outflow is estimated to be in the range of 1.5–4 nmol mol$^{-1}$ day$^{-1}$ (Fig. 6). Note that the ozone mixing ratio is reported to be ~30±2 nmol mol$^{-1}$ during July 2009 over Ananthapur, a rural site in central India, indicating the enhancement due to regional ozone production (Fig. 6). As shown in Table 2, while average $O_3$ mixing ratios over the west coast of India and the Arabian Sea are in the range of 9–25 nmol mol$^{-1}$ during the monsoon season, the average $O_3$ mixing ratio is ~ 30 nmol mol$^{-1}$ over the central Indian station and the BoB.

As shown in Fig. 6, during the cruise observations, $O_3$ mixing ratios were 27±3 and 28±5 nmol mol$^{-1}$ for July 21, 2009 and August 15, 2009, for which back-trajectories crossed Thiruvananthapuram on July 20, 2009 and August 13, 2009, with daytime $O_3$ values of 23±6 and 25±6 nmol mol$^{-1}$, respectively. The difference of 3–4 nmol mol$^{-1}$ between ozone mixing ratios over the BoB and Thurivannthapuram represents the en route photochemical production of ozone in the airmasses toward the observation points over the BoB. It is further found that the airmasses were typically below 700 meters, and generally within the marine boundary layer (e.g. mean boundary layer height ~897 m during winter over the BoB; Subrahamanyam et al., 2012). The enhancements in $O_3$ are attributed here to in situ photochemical build-up while moving towards the BoB, which has been noted in previous experiments and model simulations (e.g. Lal and Lawrence, 2001; Ojha et al., 2012).

CO showed a sharp enhancement (denoted with red arrows in Fig. 5b) on August 7 and 11, 2009, coinciding with a longer residence time over the Indian region. Figure 7 shows backward airmass trajectories above the measurement locations, along with the distribution of anthropogenic CO emissions from the INTEX-B inventory, representative of the year 2006. The airmasses over the BoB are found to be influenced by emission hotspots (corresponding emission of 250–350 mol km$^{-2}$ hr$^{-1}$). The airmasses took about half a day to be transported from the emission hotspot to the observation location over the BoB. CO mixing ratios measured at Bhubaneswar (20.30° N; 85.83° E), a station within the hotspot region, is ~251±58 nmol mol$^{-1}$ during the monsoon season (June–August 2011–2012; Mahapatra et al., 2014), with the elevated CO emissions in the Bhubaneswar region being attributed to industrial activities. The higher CO mixing ratios ~200 nmol mol$^{-1}$ is inline with the monsoonal values observaed at Bhubaneswar. The CO mixing ratios around 150 nmol mol$^{-1}$ were sampled on August 11, 2009 near the coastal source regions.

Additionally, CO mixing ratios over central BoB (101 nmol mol$^{-1}$) were only slightly higher than those over northern BoB (95 nmol mol$^{-1}$). We suggest that this is partially due to higher emissions over southern India, against the shorter residence of airmasses over land and the relatively longer lifetime of CO.

### 5.2 WRF-Chem simulations



WRF-Chem simulations, as described in Section 4, are used to evaluate the performance of the model in reproducing our measurements, and to investigate the underlying processes that caused the observed variabilities in $O_3$ and CO. Before evaluating the specific chemical species, variations in the meteorological parameters simulated by the model are briefly evaluated. Figure 8 compares WRF-Chem simulations with the in situ measurements of meteorological parameters along the cruise track. Overall, WRF-Chem reproduces these meteorological parameters with only small

mean biases, such as -2.3 hPa in pressure, -0.5° C in temperature, and -1.4% in relative humidity (Table 3). However, model shows limitations in capturing some of the sharp reductions in air temperature and associated enhancements in relative humidity.

Figure 9 compares WRF-Chem-simulated $O_3$ and CO with in situ measurements taken along the cruise track. WRF-Chem is found to reproduce the observed variations in $O_3$ and CO over the BoB during the summer monsoon season

with an overestimation of absolute $O_3$ levels by 6.2 nmol mol$^{-1}$ (i.e. ~21% of averaged $O_3$ value, 29.7 nmol mol$^{-1}$) and absolute CO levels by 22 nmol mol$^{-1}$ (i.e. ~23% of averaged CO value, 96 nmol mol$^{-1}$). It should be noted here that the average CO mixing ratio of 96 nmol mol$^{-1}$ is slightly higher than its actual value, as data points below the detection limit of the instrument are discarded. Biases in the model simulations can be attributed to the uncertainties in the simulated meteorology and input emissions; however, in the present study, we use the model fields mainly to

investigate temporal variations. The squared correlation coefficients between daily averaged in situ measured and simulated $O_3$ and CO are 0.67 and 0.19, respectively. The higher value of the squared correlation coefficient for $O_3$ demonstrates WRF-Chem's ability to reproduce the observed broad features in surface $O_3$ over the BoB. Note that the sharp reductions that caused very low ozone during rainfall episodes are not captured by WRF-Chem, except during the event of August 10–11, 2009. This will be discussed in detail in Section 5.4.


### 5.3    Diurnal variation

Figure 10 shows the mean delta-diurnal variation, that is the mean value subtracted from the mean diurnal pattern, in surface $O_3$ from observations and model simulations for the location where the ship was kept stationary (89° E, 19° N) for a period of about 15 days. Ozone at each hour shown here is an average of 10 to 15 observations. Ship

exhaust contaminated the observations for a period of time between 5 to 14 hours long; data corresponding to this period is therefore discarded from analysis, leading to a gap. Both the WRF-Chem model and observations showed only small variability from mean values (delta ozone = -2 to +2 nmol mol$^{-1}$) during the summer monsoon season. Neither our limited measurements nor the simulations exhibited any tendency towards net photochemical build-up in ozone after sunrise during the monsoon. The observations available during 5 to 14 hours on July 23 and 24, 2009

also do not show any daytime build up. A net daytime photochemical build up has been reported over the BoB during both pre-monsoon (Nair et al., 2011) and post-monsoon season (Mallik et al., 2013), as shown for comparison in Figure 10. The absence of net day-time photochemical build-up and the highly correlated variability of ozone with residence time over the Indian region (Section 5.1) suggest that spatio-temporal variations in surface ozone over the BoB during monsoon season are associated with the direct transport, supplemented with en route

photochemistry. Note that, due to the insufficient number of observations, diurnal variations in CO and $CH_4$ are not discussed.





### 5.4    Ozone variations during rainfall events

An interesting phenomenon observed during the CTCZ experiment is the abrupt reduction in ozone mixing ratios
that accompanied the onset of heavy rainfall, despite the low solubility of ozone in water. In this section we
investigate the possible causes of these low-ozone events during rainfall over the BoB.

Figure 11 shows variations in $O_3$ (black circles), TRMM-retrieved rainfall (thick grey vertical bars), and WRF-Chem-simulated vertical winds at pressure levels ranging from 950–750 hPa (coloured bars) during four such events
on July 21, 26, and 28–29 and on August 10–11, 2009. As high time-resolution in situ measurements of rainfall were
not available aboard ship, Figure 11 therefore uses 3-hourly rainfall retrievals from the TRMM, co-located with
ozone measurements. During these events, CO mixing ratios also show a reduction of about ~56 nmol mol$^{-1}$, with
observed values falling below the detection limit of the instrument during the first event of July 21, 2009 (not
shown). Although CO measurements are not available for the second and third event, during the fourth event
(August 10–11, 2009) CO mixing ratios showed an enhancement due to transport from strong source regions (see
Section 5.1). While the first three low-$O_3$ events are not captured by WRF-Chem (Fig 9a), the fourth event is
reproduced.

Wet scavenging does not directly reduce ozone, as its water solubility is low; as a result, some dynamic process
could be responsible for the observed reductions in ozone during rainfall. Airmasses could undergo downdrafts
during heavy rainfall (Kumar et al., 2005) through air drag by the falling rain drops and in mesoscale subsidence that
compensates convective updrafts. We suggest that, in the presence of ozone-poor airmass aloft, a downdraft would
result in reductions in surface ozone mixing ratios. The model-simulated meteorology shows occurrences of
downdrafts at different pressure levels during the first three events on July 21, 26, and 28–29 (Fig. 11a–c), which we
further corroborate with measurements of air temperature aboard. Downdrafts of free tropospheric air could lead to a
reduction in near-surface temperature by as much as 10 °C within a few minutes (Ahrens, 2009). Air temperature
measured aboard ship showed a sharp decrease of 2–4 °C, coinciding with the first three low-ozone events (Fig. 12).
The reductions in temperature caused by downdrafts are generally short-lived (Ahrens, 2009), it is confirmed in the
case of these events (Fig. 12).

Model-simulated vertical winds and variations in air temperature suggest that downdrafts did occur during the first
three rainfall events. As in situ measurements of ozone vertical profiles are not available over the BoB during the
summer monsoon season, we instead used observations taken at Thumba, Thiruvananthapuram, in the southern
Indian region as a case study to investigate this hypothesis. For general details of the typical diurnal and seasonal
variations in $O_3$ at Thumba, please see Nair et al. (2002), David and Nair (2011), and Girach et al. (2012). Figure
13a shows the temporal variation in surface $O_3$ on July 15, 2011 at Thumba, along with 5-minute accumulated
rainfall. Here, surface ozone is observed to decline from 25 to 13 nmol mol$^{-1}$ within 15–20 minutes, coinciding with
the occurrence of intense rainfall (3.5–0.5 mm rain over a period of 5 minutes). Measurements of the $O_3$ vertical
profile are not available for this day due to the rainy conditions; a profile measured on July 28, 2011 is therefore
shown in Fig. 13b. This profile has significantly lower ozone mixing ratios aloft (~22 nmol mol$^{-1}$ at ~1 km) than
near surface (~42 nmol mol$^{-1}$). A typical mixed layer height is about 0.6±0.2 km over Thumba, Thiruvananthapuram





(Nair et al., 2011) during the summer monsoon season; above this height, $O_3$ mixing ratios sharply decrease with altitude. The present case study suggests the presence of ozone-poor airmasses aloft than those near the surface over the south Indian region during summer monsoon. With an ozone distribution as observed in the present case study at Thumba, the downdraft during intense rainfall could lead to the mixing of free-tropospheric air with near-surface air, or to the replacement of surface air with free-tropospheric ozone-poor air.

Although air temperature measurements could not be made during the fourth event (10–11 August 2009) due to a technical problem, model meteorology does not indicate a downdraft during this event (Fig. 11d), indicating the dominance of a different process. As WRF-Chem-simulated ozone variability is in good agreement with observations during this event, we used various tendency terms from WRF-Chem to investigate the relative influences of different processes. The variations in instantaneous values for horizontal advection tendency, vertical advection tendency, and net tendency (i.e. the sum of chemical, vertical mixing, convective, vertical advection, and horizontal advection), along with modelled $O_3$ over the two locations during the event are shown in Figure 14. The tendency values shown here are derived by subtracting the accumulated tendencies at $(n-1)^{th}$ hour from the accumulated tendency at $n^{th}$ hour. The vertical dotted lines show the time of a low-ozone event.

Both the horizontal and net tendencies (Fig. 14b,e) show negative values, indicating that they are contributing towards a reduction in ozone mixing ratios (Fig. 14a,d). However, as the time of the event approaches, it is the horizontal advection tendency term that is significantly negative (Fig. 14c, f), while other terms are small and close to zero. Horizontal advection is therefore suggested to dominate during the low-ozone event of August 10–11, 2009. The influence of horizontal advection on $O_3$ during this event is shown more clearly in Fig. 15, which shows the spatial distribution of $O_3$ and CO from WRF-Chem before the event (16:00 and 19:00 UT) and during the event (22:00 UT) on August 10, 2009. The white triangles show the two locations where the event was observed. During 16:00 and 19:00 UT, a patch of high ozone mixing ratios (45 nmol mol$^{-1}$ and higher) is seen to be distributed over a large region surrounding the measurement location. This large patch of elevated ozone mixing ratios is horizontally advected eastward from 16:00 to 19:00 and then towards 22:00 UT (event time). As a result of this rapid advection, the high-ozone airmasses are transported from the coastal regions to deeper into the BoB; by the time they reached the location of observation, ozone mixing ratios are observed to be lower (25–35 nmol mol$^{-1}$) during the event time (22:00 UT). A patch of higher levels of CO (~300 nmol mol$^{-1}$) was also found to be distributed across the east coast of the Indian region. Transport and dilution of this CO patch is, however, less pronounced than the high-ozone airmasses, possibly due to the relatively longer lifetime of CO. Thus, we suggest that the horizontal advection played a key role in transporting $O_3$-rich airmasses deeper into the BoB region, while it diluted $O_3$ levels near the coastal regions in southern India during the fourth event.

### 5.5 Seasonal variation in trace gases over the BoB

In this section, we combine the first monsoon-time measurements of ozone taken in the present study, with data from previous campaigns (see Table 3) to investigate the seasonal variation in ozone over the BoB (Fig. 16). $O_3$, CO, and $CH_4$ mixing ratios are averaged over northern and central BoB regions, as defined in Fig. 3.


Overall, higher $O_3$ mixing ratios are present over both northern and central BoB during the winter, while significantly lower $O_3$ levels are observed during the spring–summer (with more scatter in the data over central BoB). The $O_3$ seasonal amplitude (i.e., the range from maxima to minima) is estimated to be ~ 39 nmol mol$^{-1}$ over northern BoB and ~ 27 nmol mol$^{-1}$ over central BoB. The monsoonal surface ozone mixing ratios (~30±7 nmol mol$^{-1}$) are nearly half those observed during winter (63±5 nmol mol$^{-1}$) over northern BoB. During December 2008–

January 2009, February 2003, March 2006, and November 2010, the ozone mixing ratios were significantly higher (by ~3–22 nmol mol$^{-1}$) over northern BoB than those over central BoB. However, over the course of February 2001, ozone mixing ratios were higher over central BoB (~38 nmol mol$^{-1}$) than that over northern BoB (~14 nmol mol$^{-1}$). In contrast, during summer monsoon season, average ozone mixing ratios are comparable or only slightly higher over northern BoB (30±7 nmol mol$^{-1}$) as compared to that over central BoB (27±5 nmol mol$^{-1}$).

As compared with the summer monsoon season, when CO mixing ratios were lower, over northern BoB, CO mixing ratios were higher during the winter, while over central BoB, CO mixing ratios were higher during the pre-monsoon season. For $O_3$, spring–summer had the lower mixing ratios in both regions. The seasonal amplitude in CO mixing ratios is estimated to be ~205 nmol mol$^{-1}$ over northern BoB and ~124 nmol mol$^{-1}$ over central BoB. The monsoonal CO mixing ratio (~95 nmol mol$^{-1}$) is about one third that of the winter season (302 nmol mol$^{-1}$) over northern BoB.

During the present study, average CO mixing ratios were comparable over northern (95±25 nmol mol$^{-1}$) and central BoB (101±27 nmol mol$^{-1}$).

A clear inference about seasonal patterns is difficult in the case of $CH_4$, however a tendency of lower levels towards winter can be seen. Higher mixing ratios ~1.95 (~1.91) µmol mol$^{-1}$ were observed during November 2010 over northern BoB, and during February–March 2001 over central BoB, as compared to those from other studies. In the

present study, average mixing ratios of methane are significantly higher over northern BoB (1.86±0.12 µmol mol$^{-1}$) as compared to over central BoB (1.72±0.14 µmol mol$^{-1}$) during the summer monsoon season. The higher tropospheric $CH_4$ that has been observed over the central and northern Indian landmass during the summer monsoon season (Kavitha and Nair, 2016) could be responsible for the higher $CH_4$ that is observed over northern BoB in the present study. Owing to the longer lifetime of $CH_4$, diffusion of $CH_4$ from a hotspot region over the eastern IGP to

northern BoB might be the other source of higher $CH_4$ levels over northern BoB during summer monsoon season. An analysis of an emission inventory by sector over the hotspot region (i.e. eastern IGP) indicates that these higher methane emissions are due to rice cultivation, waste treatment and livestock. The surface $CH_4$ observations obtained during the present study show the highest variability (i.e. the difference between maxima and minima) when compared to earlier studies: 0.53 µmol mol$^{-1}$ over northern BoB and 0.39 µmol mol$^{-1}$ over central BoB. We attribute

this high variability to the relative source strengths over central and northern India as compared to southern India, highlighting the regional differences in $CH_4$ variability across India (Kavitha and Nair, 2016).

Seasonal variations in trace gases over the BoB are attributed to seasonal changes in the meteorological conditions, emissions, and photochemistry over the South Asian region, as well as to synoptic scale transport patterns. Wintertime stronger westerly winds transport the pollution from South Asia including that of the Indo-Gangetic

basin to the BoB region. Monsoonal circulation, in contrast, carries cleaner marine airmasses to the BoB from the oceanic regions. However, as observed during the CTCZ, polluted continental or coastal airmasses can also




occasionally be transported deeper over the BoB. Intense monsoonal rainfall generally leads to wet removal of ozone precursors, while cloudy and rainy meteorological conditions suppress ozone formation. Along with the importance of monsoonal convection in cloud formation, rainfall, and uplifting the boundary layer pollution, rapid

horizontal advection is also an important process during the summer monsoon, especially affecting the near-surface variability of trace gases over the oceanic regions adjacent to India.

### 6. Conclusions

In this paper, we presented the ship-borne in situ measurements of $O_3$, CO, and $CH_4$ that were carried out as a part of

the CTCZ experiment over the BoB during July–August 2009, the first time that such measurements had been taken over this region during the summer monsoon season. We analyzed the spatial and temporal variations in our observations and compared them with simulations from a regional chemistry transport model (WRF-Chem), as well as with observations from previous campaigns over the BoB. The main conclusions from the study are:

1. These first monsoonal observations of $O_3$, CO, and $CH_4$ show significant spatio-temporal variability over the

BoB, with mixing ratios varying in the range of 8–54 (mean: 29.7±6.8) nmol mol$^{-1}$, 50–200 (mean: 96±25) nmol mol$^{-1}$, and 1.57–2.15 (mean: 1.83±0.14) μmol mol$^{-1}$, respectively. The $O_3$, CO, and $CH_4$ mixing ratios are slightly higher or comparable ($O_3$: 30±7 nmol mol$^{-1}$, CO: 95±25 nmol mol$^{-1}$, $CH_4$: 1.86±0.12 μmol mol$^{-1}$) over northern BoB as compared to over central BoB ($O_3$: 27±5 nmol mol$^{-1}$, CO: 101±27 nmol mol$^{-1}$, $CH_4$: 1.72±0.14 μmol mol$^{-1}$). The difference (~0.14 μmol mol$^{-1}$) between $CH_4$ mixing ratios over northern and central BoB is

most significant.

2. Back-trajectory analysis shows effects of long-range transport from northern or central India to northern BoB, and from southern India to central BoB. The correlated variations of $O_3$ mixing ratio and percentage residence time of air parcels over the Indian regions suggest that the enrichment of ozone and precursors in air parcels over the BoB is associated with both emissions and photochemistry over the Indian region. The trajectory analysis

and mean diurnal variations show that the observed variation in surface $O_3$ is primarily due to transport and en route photochemistry, rather than to local photochemical production over the BoB during monsoon season.

3. The observed spatio-temporal variations in surface $O_3$ and CO during summer monsoon season are generally reproduced by the WRF-Chem model, although the absolute mixing ratios of $O_3$ and CO are typically overestimated by about 20%.

4. We observed four low-ozone events coinciding with intense rainfall over the BoB. After analysing the observed variability in air temperature, model simulations of vertical winds, and an ozone-profile case study from southern India, we suggest that first three low-ozone events were due to strong downdrafts of ozone-poor airmasses. Analysis of the fourth low-ozone event, which is successfully reproduced by the model, shows the pivotal role of horizontal advection in transporting ozone-rich airmasses deeper over the BoB.

5. Finally, we combined our monsoon time measurements with previous campaigns over the BoB during other seasons to investigate the seasonal variability in trace gases over the BoB. $O_3$ and CO are shown to have pronounced seasonality, $O_3$ having amplitudes of about 39 and 27 nmol mol$^{-1}$, and CO having amplitudes of about 207 and 124 nmol mol$^{-1}$ over northern and central BoB, respectively.



Our study fills a gap of experimental data during the summer monsoon over the BoB, providing information on the
extent of seasonal variability. We recommend supplementing these findings with ship-borne experiments featuring
collocated vertical profile observations from balloon-borne and aircraft-based platforms over the oceanic regions
surrounding India to better understand the role of both large-scale dynamics (e.g. Ojha et al., 2016) and of regional
influences due to South Asian outflow (see Lawrence and Lelieveld, 2010, and references therein). Such a future
study would also improve our understanding of the changes that take place in the atmospheric oxidation capacity
during the summer monsoon season.

**Acknowledgements**

We thank all of the CTCZ and ICRP organizers for providing the opportunity to participate in the 2009 CTCZ
experiment. We are thankful to the Director of the National Centre for Antarctic and Ocean Research (NCAOR),
Goa for providing ship-board facilities. We gratefully acknowledge Prof. G. S. Bhatt (Indian Institute of Science,
Bengaluru, India) and his team for providing the measurements of meteorological parameters onboard ship. We also
thank the chief scientist on board *SagarKanya* for providing necessary support during the cruise. The authors
gratefully acknowledge the NOAA Air Resources Laboratory (ARL) for the providing the HYSPLIT transport and
dispersion model and READY website (http://www.arl.noaa.gov/ready.php) used in this publication. The rainfall
estimations (3B42) from the TRMM satellite were obtained from the NASA/GSFC via their website
http://mirador.gsfc.nasa.gov/.          Use        of         INTEX-B        and        HTAP
(http://edgar.jrc.ec.europa.eu/htap_v2/index.php?SECURE=123)        anthropogenic      emissions     is    gratefully
acknowledged. Initial and boundary-conditions data for meteorological fields were used from the ERA interim of
ECMWF. Use of MOZART-4/GEOS5 initial and boundary conditions data for chemical fields is acknowledged.
Data/processors for anthropogenic emissions, biogenic emissions, and biomass burning obtained from NCAR ACD
website are gratefully acknowledged. The authors acknowledge the use of MPG supercomputer HYDRA
(http://www.mpcdf.mpg.de/services/computing/hydra) for model simulations.

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





**Tables**

**Table 1.** The WRF-Chem options used for parameterization of atmospheric processes.

| Atmospheric Process | Option used |
|---|---|
| Cloud microphysics | Lin et al. scheme (Lin et al., 1983) |
| Longwave radiation | Rapid Radiative Transfer Model (RRTM; Mlawer et al., 1997) |
| Shortwave radiation | Goddard shortwave scheme (Chou and Suarez, 1994) |
| Surface Layer | Monin–Obukhov scheme (Janjic, 1996) |
| Land surface option | Noah Land Surface Model (Chen and Dudhia, 2001) |
| Urban surface physics | Urban Canopy Model |
| Planetary boundary layer | Mellor–Yamada–Janjic scheme (Janjic, 2002) |
| Cumulus parameterization | New Grell scheme (G3) |




**Table 2.** A comparison of averaged surface $O_3$ mixing ratios over various sites. [*]boundary layer ozone over the Arabian Sea.

| Observation site | Longitude (° E) | Latitude (° N) | Observation period during monsoon season | Mean Surface Daytime Ozone | Standard Deviation | Reference |
|---|---|---|---|---|---|---|
| Arabian Sea | | | | | | |
| Arabian Sea | 69 –76 | 9 –19 | July–August 2002 | 9 | - | Ali et al., 2009 |
| Ahmedabad | 72.6 | 23 | July–August 2003–2007 | 25[*] | - | Srivastava et al., 2012 |
| Western coast of India | | | | | | |
| Thiruvananthapuram | 76.9 | 8.5 | August 2009 | 23 | 7 | Present Study |
| Thiruvananthapuram | 76.9 | 8.5 | June–August 2008 | 19 | 6 | David and Nair, 2011 |
| Kannur | 75.4 | 11.9 | July 2010–2011 | 11 | 4 | Nishanth et al., 2014 |
| MtAbu (1.6km amsl) | 72.7 | 24.6 | August 1993–2000 | 25 | 9 | Naja et al., 2003 |
| Ahmedabad | 72.6 | 23 | July 1991–1995 | 22 | 8 | Lal et al., 2002 |
| Ahmedabad | 72.6 | 23 | August 1991–1995 | 17 | 4 | Lal et al., 2002 |
| Central India | | | | | | |
| Anantpur | 77.65 | 14.62 | July 2009 | 30 | 2 | Reddy et al., 2011 |
| Eastern coast of India | | | | | | |
| Bhubaneswar | 86.4 | 20.5 | June–August 2011–2012 | 29 | 6 | Mahapatra et al., 2014 |
| Bay of Bengal | | | | | | |



| Bay of Bengal | 80.3–90.1 | 11–21.1 | July–August 2009 | 30 | 7 | Present Study |
|---|---|---|---|---|---|---|


**Table 3.** A comparison of mean values from observations with model-simulated parameters along with the mean bias. The squared correlation coefficients correspond to the linear regression analysis between daily averaged in situ and simulated parameters.

| Parameter | Observation | Model (WRF-Chem) | Mean bias | $R^2$ |
|---|---|---|---|---|
| Pressure (hPa) | 1001.3±2.1 | 999.0±2.4 | -2.3 | 0.93 |
| Temperature (°C) | 29.3±0.9 | 28.8±0.6 | -0.5 | 0.12 |
| Relative Humidity (%) | 87.9±4.2 | 86.5±2.8 | -1.4 | 0.54 |
| $O_3$ (nmol mol$^{-1}$) | 29.7±6.8 | 35.9±8.3 | 6.2 | 0.67 |
| CO (nmol mol$^{-1}$) | 96±25 | 118±37 | 22 | 0.19 |




**Table 4**. A comparison of average mixing ratios of surface trace gases measured over northern BoB (81-91° E, 16-21.5° N) and central BoB (80-91° E, 11-16° N) in different seasons as measured during different experiments. The range of mixing ratios (i.e. minima–maxima) is given in the brackets. *CO mixing ratios below the detection limit (i.e. 50 nmol mol$^{-1}$) are not considered in the analysis.


| Study period | Name of Experiment | Reference | $O_3$ (nmol mol$^{-1}$) over northern BoB | $O_3$ (nmol mol$^{-1}$) over central BoB | CO (nmol mol$^{-1}$) over northern BoB | CO (nmol mol$^{-1}$) over central BoB | $CH_4$ (μmol mol$^{-1}$) over northern BoB | $CH_4$ (μmol mol$^{-1}$) over central BoB |
|---|---|---|---|---|---|---|---|---|
| December 2008– January 2009 | W_ICARB | David et al., 2011 | 63.0±4.7 (50.8–73.8) | 40.9±6.7 (27.7–63.5) | 302±68 (140–450) | 188±53 (50–320) | No data | No data |
| February 2003 | BOBEX-II | Lal et al., 2007 | ~34.1 (15.8–50.4) | ~26.8 (13.9–35.0) | ~238 (187–292) | ~192 (159–224) | ~1.77 (1.70–1.85) | ~1.73 (1.68–1.77) |
| February– March 2001 | BOBEX-I | Lal et al., 2006 | ~23.8 (16.1–38.3) | ~38.0 (19.4–62.9) | ~194 (165–235) | ~227 (97–339) | ~1.94 (1.89–2.02) | ~1.91 (1.74–2.06) |
| March– April 2006 | ICARB | Nair et al., 2011; Srivastava et al., 2012 | 27.4±2.9 (21.4–32.6) | 13.4±4.2 (3.1–24.6) | ~189 (157–235) | ~132 (96–167) | ~1.84 (1.80–1.88) | ~1.80 (1.75–1.84) |
| July–August 2009 | CTCZ | Present Study | 30.0±6.9 (8.50–54.1) | 27.5±5.0 (8.8–40.5) | 95±25* (50-198) * | 101±27* (50-157) * | 1.86 ±0.12 (1.62–2.15) | 1.72±0.14 (1.57–1.96) |
| September– October 2002 | BOBPS | Sahu et al., 2006 | ~27.3 (17.8–33.8) | ~30.6 (22.5–35.2) | ~152 (109–179) | ~141 (108–211) | ~1.79 (1.72–1.86) | ~1.73 (1.68–1.80) |
| November 2010 | No name | Mallik et al., 2013 | ~46.0 (26.7–59.6) | ~38.7 (17.8–60.8) | ~223 (131–280) | ~188 (42–266) | ~1.95 (1.85–2.06) | ~1.79 (1.67–1.93) |



**Figures**

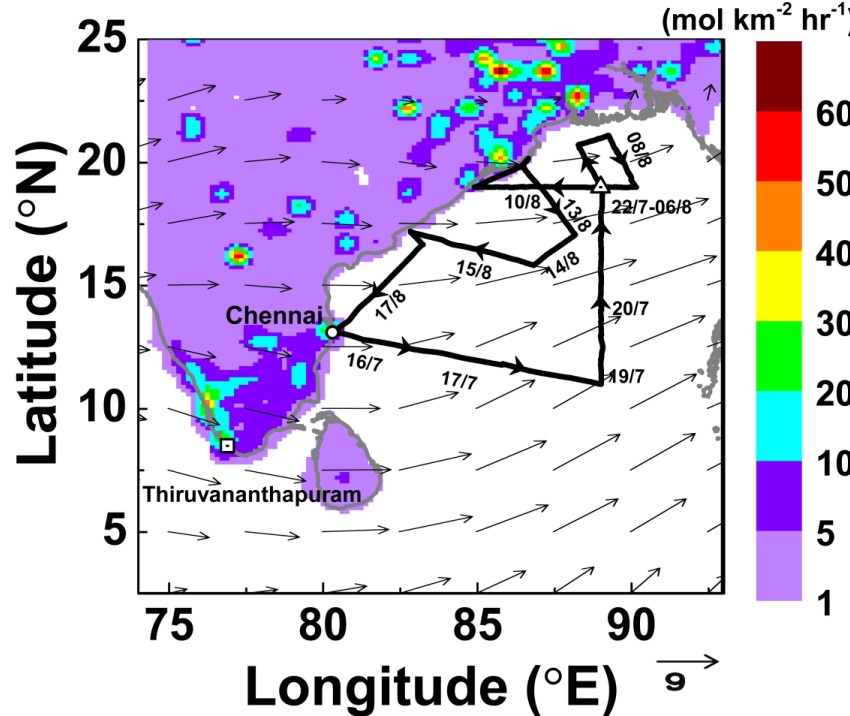

**Figure 1.** Cruise track (continuous black line) of the Research Vessel *Sagar Kanya* along with synoptic winds at 925 hPa (black thin arrows) averaged over the cruise period. Arrows marked on the track shows the ship direction. The dates corresponding to approximate ship positions are marked along the track. The circle shows the start and end point of the cruise. The square tagged with Thiruvananthapuram shows the location corresponding to the measurements shown in Fig. 15. The location at which the ship was kept stationary (July 22–August 06, 2009) is denoted with a triangle. The background colour map shows the $NO_x$ emissions over the Indian landmass for year 2006 as obtained from the INTEX-B inventory.



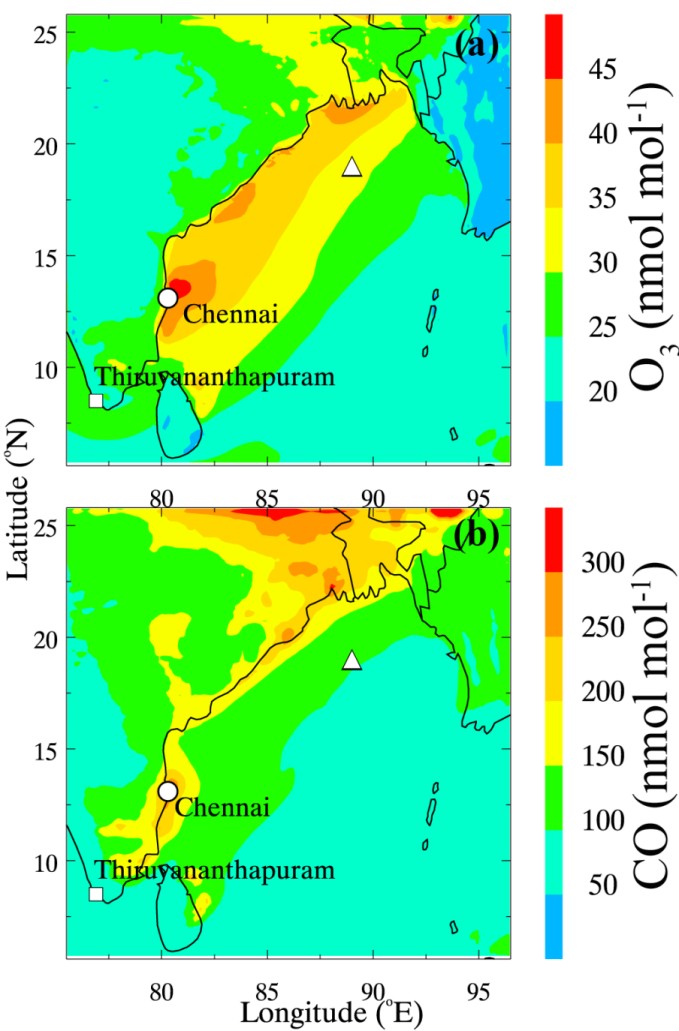

**Figure 2.** WRF-Chem-simulated spatial distribution of surface $O_3$ (a) and CO (b) averaged during the July 16–August 17, 2009 period. The location of the ship cruise start and end (Chennai), the ground-based measurement site at Thiruvananthapuram, and the location where the ship was kept stationary are shown by the white circle, square, and triangle, respectively.






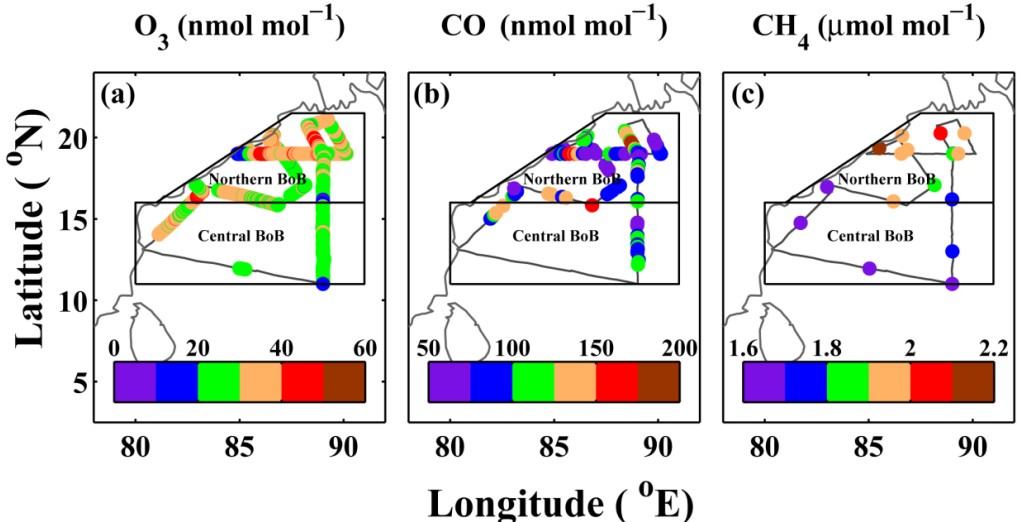

**Figure 3.** Spatial variation of surface $O_3$ (a), CO (b), and $CH_4$ (c) mixing ratios along the cruise track during the CTCZ experiment, which took place during the summer monsoon season. The solid lines demarcate the regions of central and northern BoB.






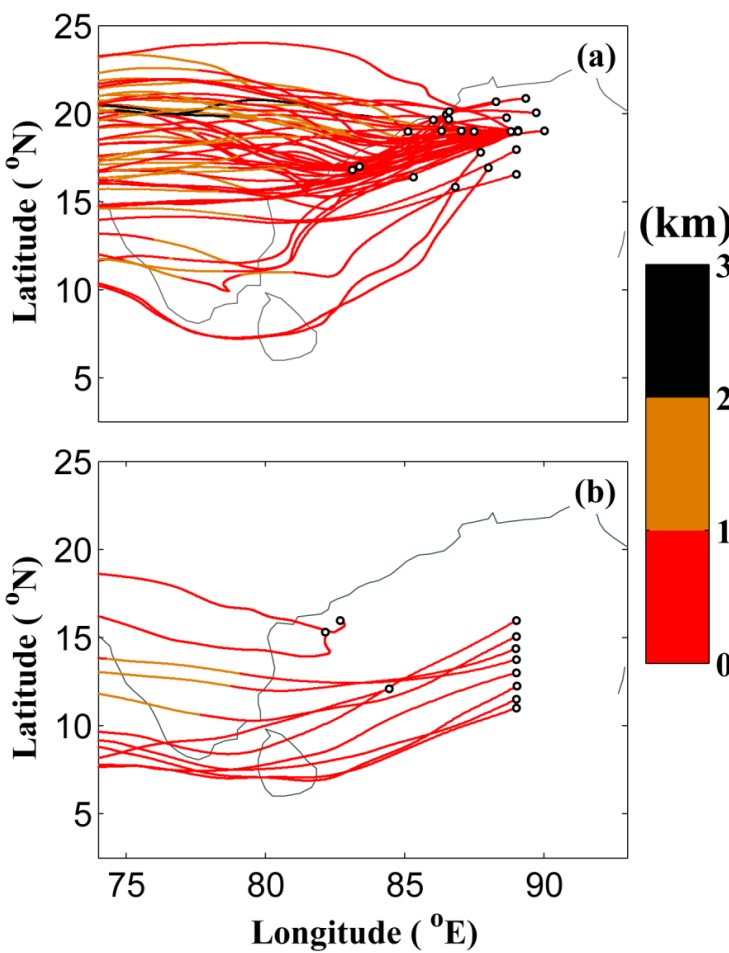

**Figure 4.** Five-day airmass back-trajectories during the study period ending at the measurement locations (small black circles) grouped for corresponding airmasses over (a) northern BoB and (b) central BoB. The colour scale shows the height (in km) of the trajectories.




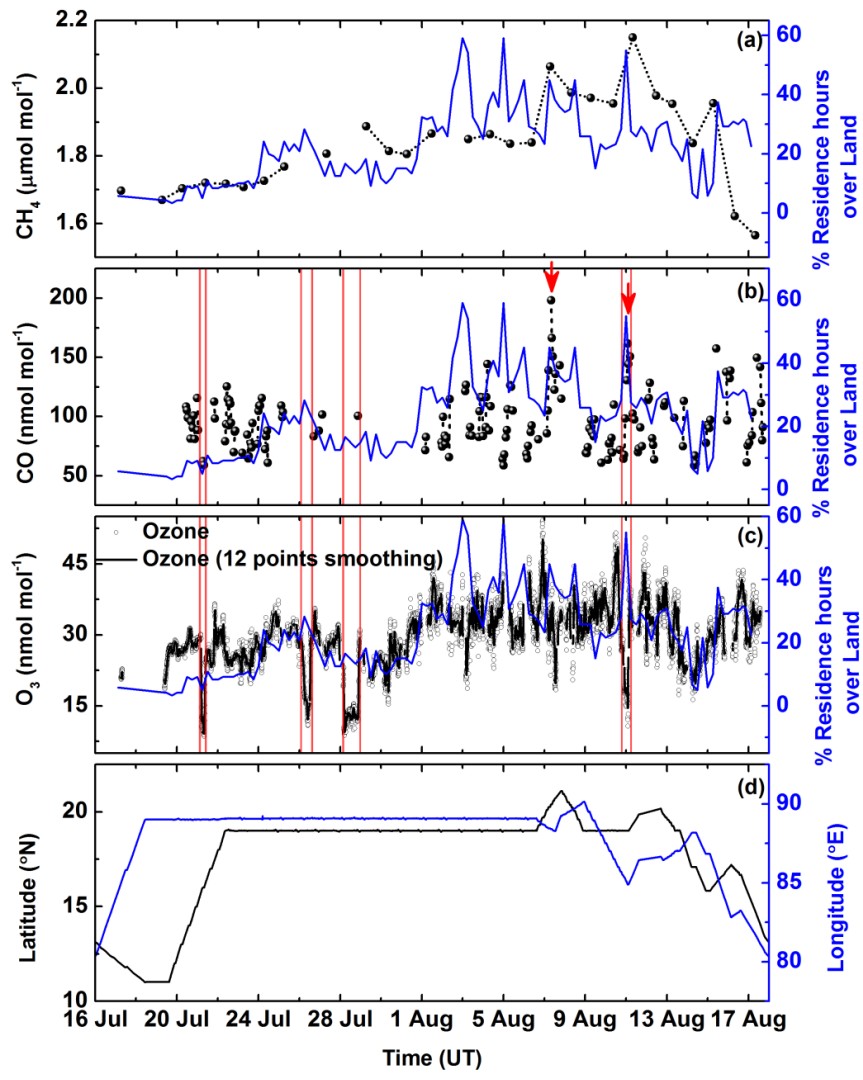

**Figure 5**. Spatio-temporal variation of surface $O_3$ (c), CO (b), and $CH_4$ (a) along with percentage residence time (blue line) over land during the summer monsoon season. The red vertical lines show the four events of sharp decrease in surface $O_3$ and CO during rainfall (Fig. 11). (d) Variations in measurement locations, latitude (black line), and longitude (blue line) corresponding to trace gas measurements shown in (a–c). Red arrows in 5b highlight elevated CO mixing ratios.



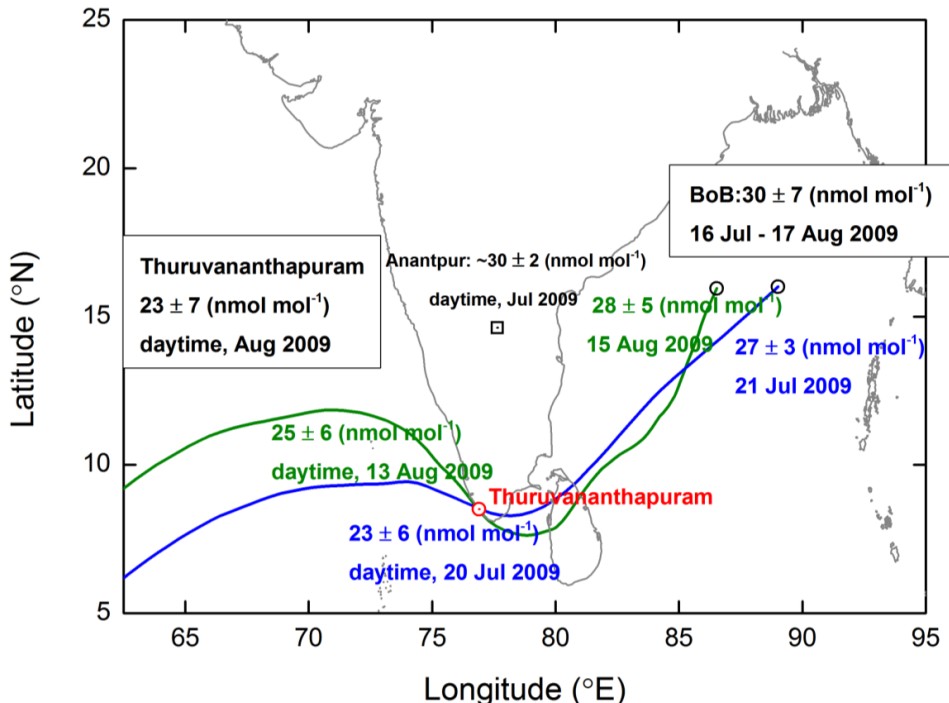

**Figure 6.** Airmass back-trajectories (blue and green curves) reaching 500 m altitude over the two locations (black
circles) of observations over the BoB for July 21 and August 15, 2009. The trajectories crossed an observational site,
Thiruvananthapuram (red circle), on July 20 and August 13, 2009. Monsoon-time average mixing ratios over
Thiruvananthapuram (which is representative of the Arabian Sea airmasses), Ananthapur (which is representative of
airmasses over the central part of southern India), and the BoB are also shown.



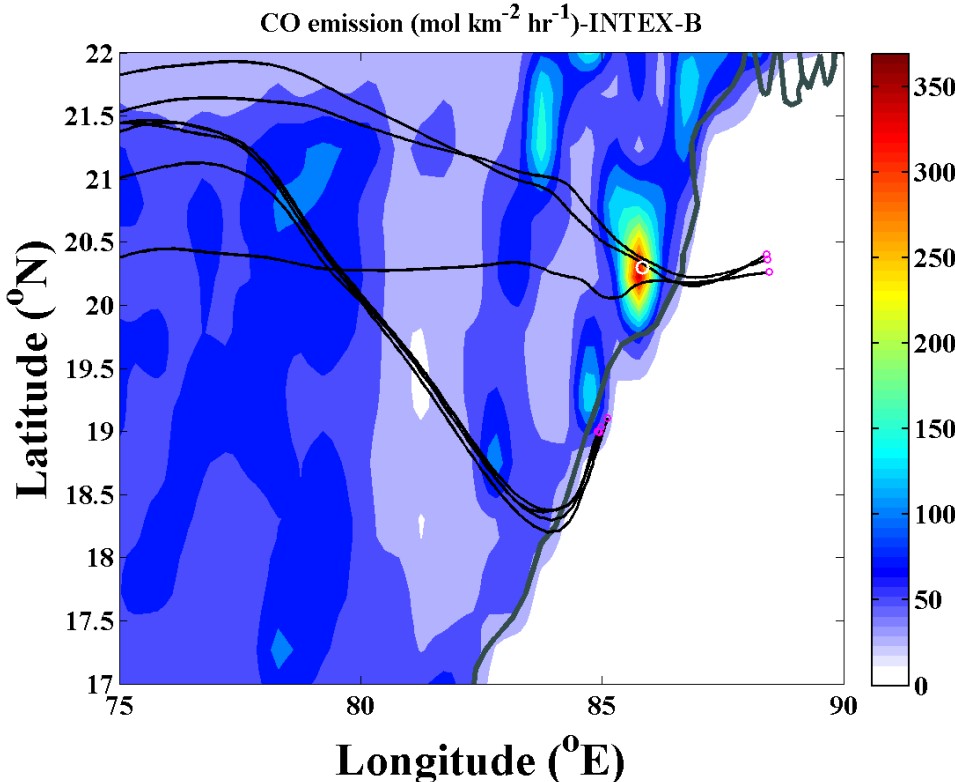

**Figure 7.** Backward Airmass trajectories (black curves) 500 m above the location of higher CO observations as marked by red arrows in Fig. 5b during August 7 and 11, 2009. The background colour map shows the spatial distribution of anthropogenic CO emissions over the Indian region for the year 2006 from INTEX-B inventory. The small circles in magenta represent the points where observations were made, as well as the end-point of trajectories. The white circle over the hot-spot region denotes an observational site, Bhubaneswar.



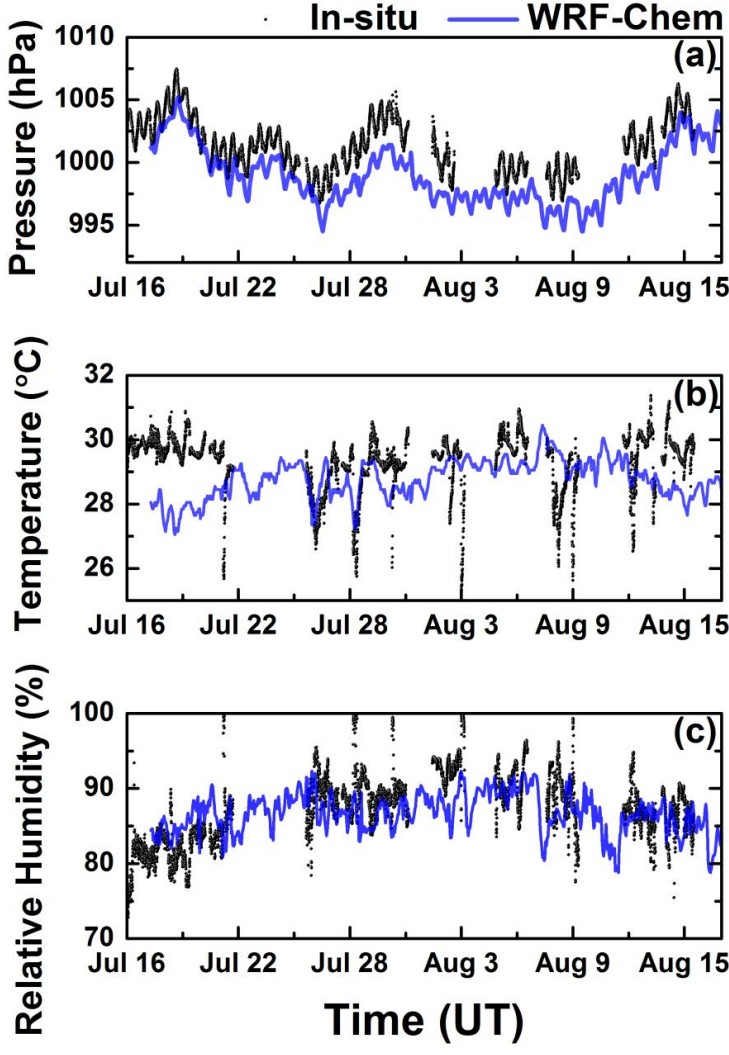

**Figure 8.** Comparison of the WRF-Chem-simulated meteorological parameters (a) pressure, (b) temperature, and (c) relative humidity with in situ measurements aboard ship during the CTCZ experiment.






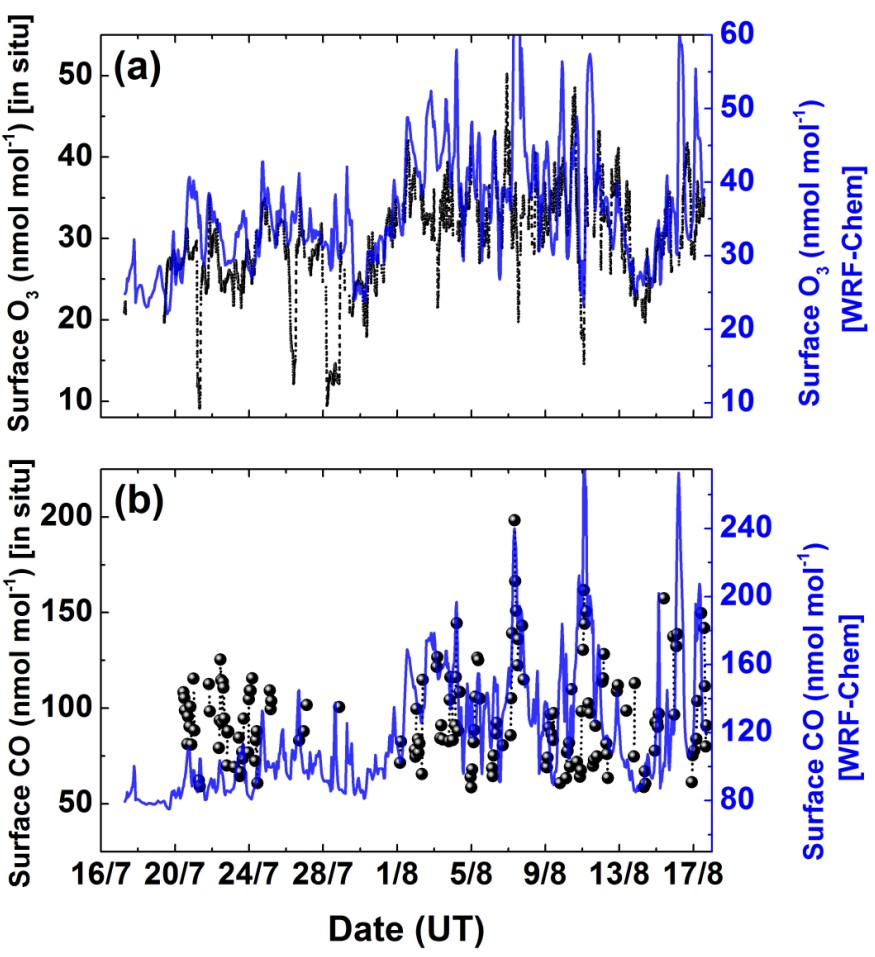

**Figure 9.** A comparison of surface $O_3$ and CO from in situ measurements (black dots) with model results from WRF-Chem (blue line) along the cruise track over the BoB during the summer monsoon season. The scale of the right axis is adjusted for WRF-Chem according to the mean biases.




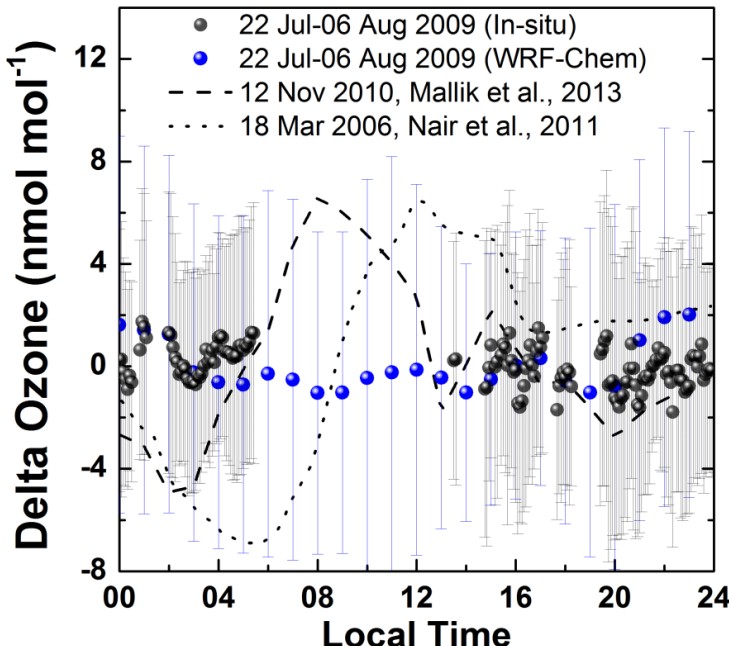

**Figure 10.** The mean delta-diurnal variation of surface ozone over the BoB at a stationary location (89° E, 19° N) along with that from WRF-Chem simulations during the summer monsoon season. Error bars represent standard deviations. The dotted and dashed curves show diurnal variations in surface O$_3$ (as adopted from Nair et al., 2011, and Mallik et al., 2013) during the pre-monsoon and post-monsoon seasons, respectively.



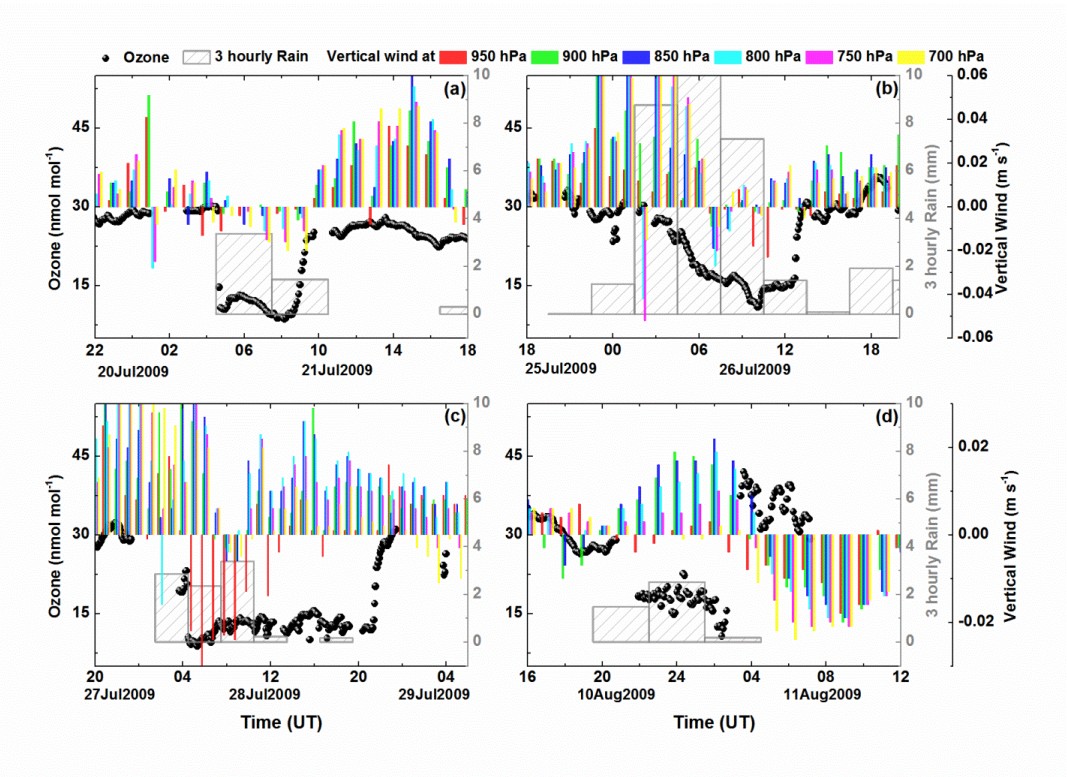

**Figure 11.** Surface $O_3$ (black dots) along with 3-hourly rainfall (grey vertical bar) during the four events of sharp decline in ozone (a–d) as marked in Fig. 5c. Colours indicate the vertical wind as simulated by WRF-Chem.





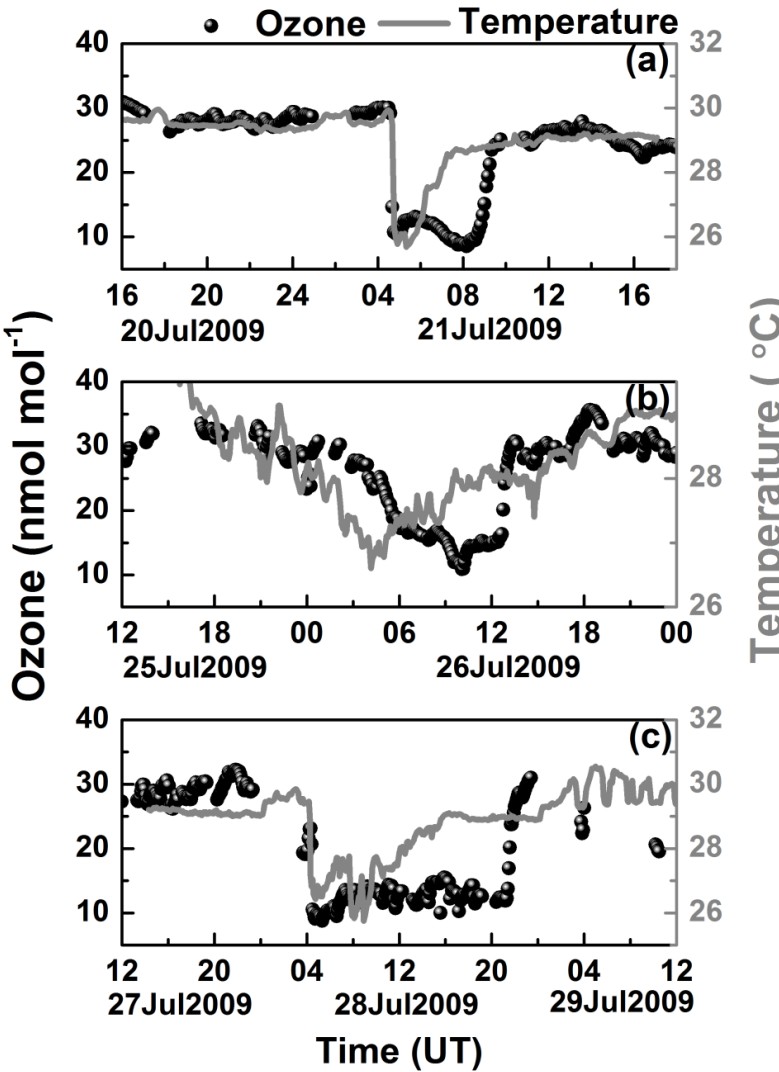

**Figure 12.** Surface O$_3$ (black dots) along with surface air temperature (grey line) during the three events of sharp decline in O$_3$ (a–c) as marked in Fig. 5.





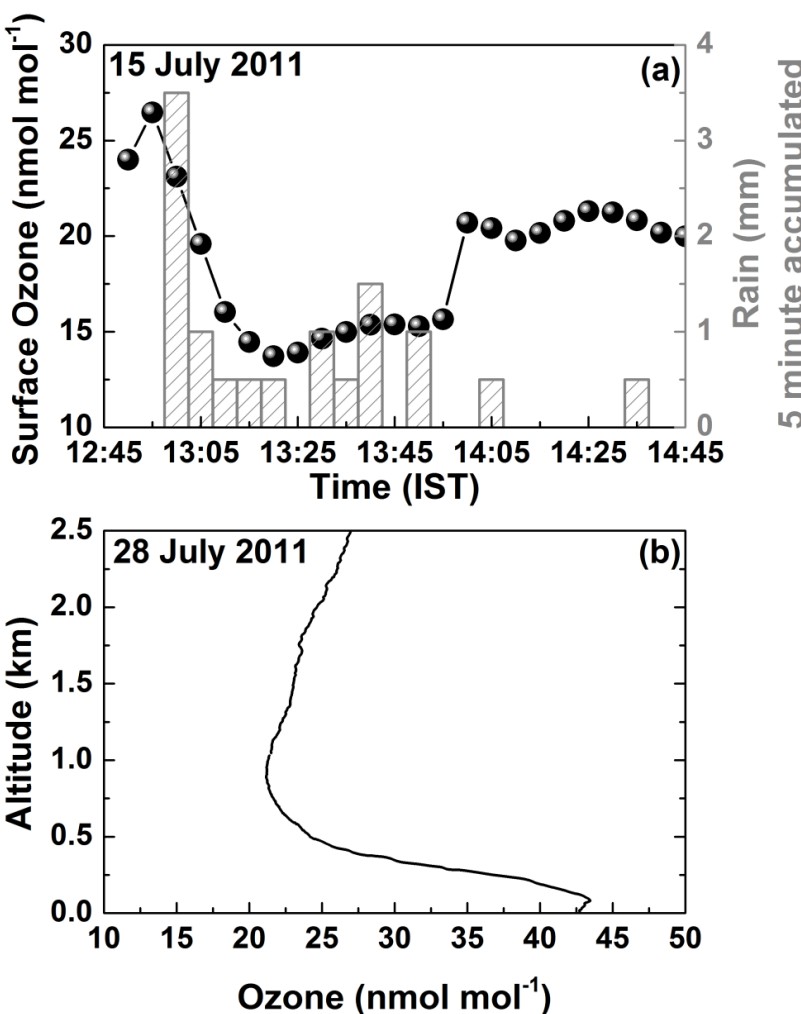


**Figure 13.** (a) Temporal variation in surface $O_3$ mixing ratio (black dots) along with 5-minute accumulated rainfall (grey vertical bars) over Thumba, Thiruvananthapuram (location of the site shown in Fig. 1 and 2) on 15 July 2011. (b) Vertical profile of $O_3$ mixing ratio over Thumba, Thiruvananthapuram as measured on 28 July 2011.



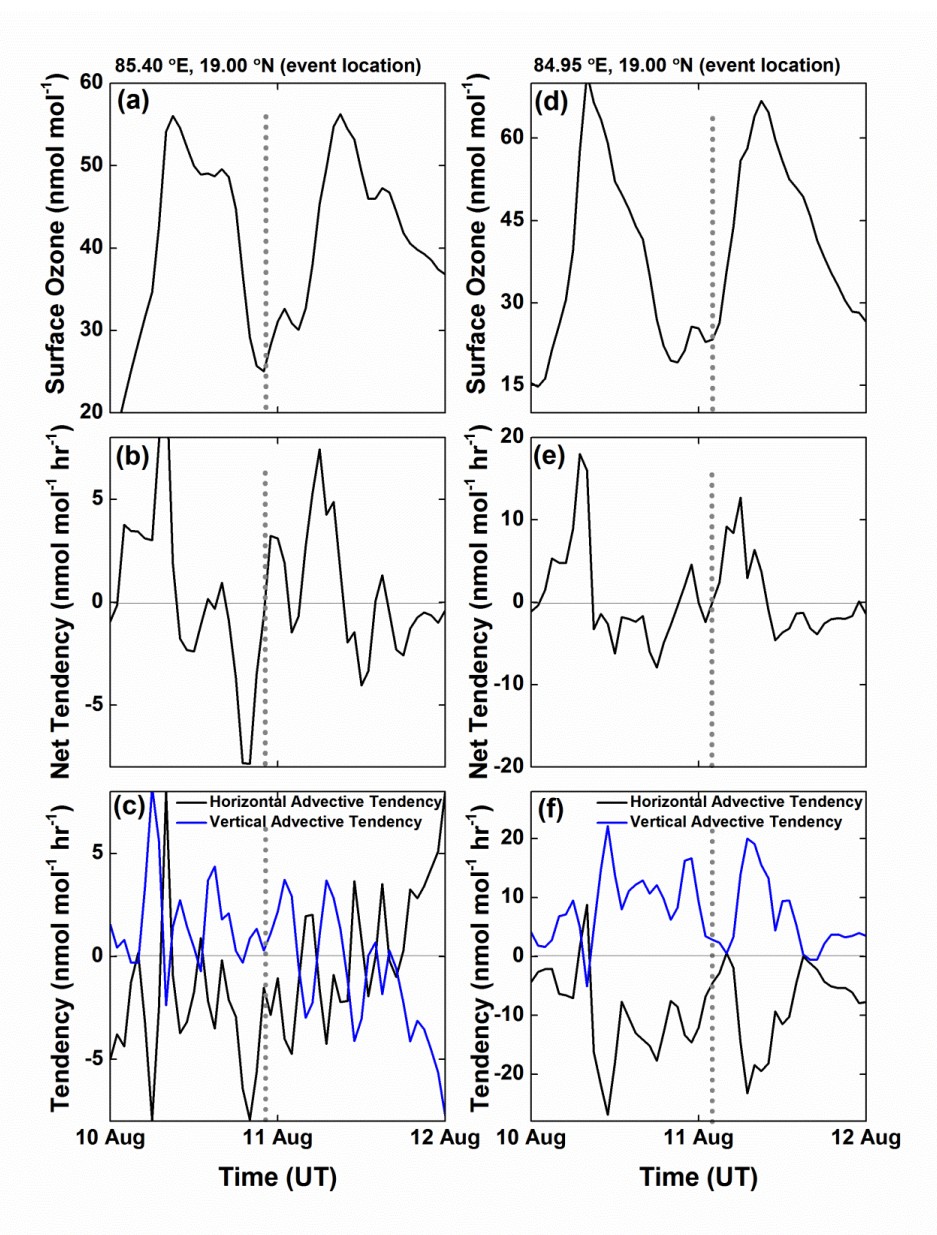


**Figure 14.** Time series of surface $O_3$ (a) and various tendency terms (b and c) over the event location during the fourth low-ozone event, as obtained from WRF-Chem simulations. 14d–f are the same as 14a–c, but for another location during the same event. These two event locations are also marked by triangles in Figure 15. Vertical dotted line shows the time of the event in the in situ observations of surface $O_3$ over the indicated locations.



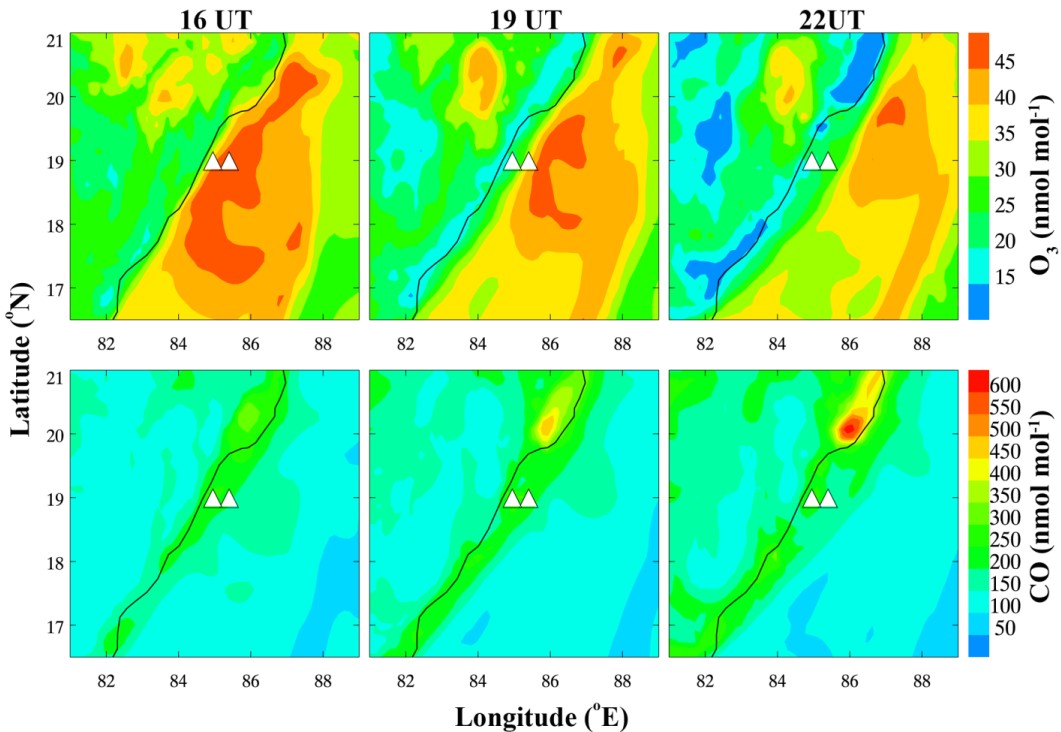


**Figure 15.** Spatial distribution of surface $O_3$ (top panel) and CO (bottom panel) at 16:00 UT and 19:00 UT on August 10, 2009, both prior to and during the fourth event, which took place 22:00 UT on August 10, 2009. White triangles show two locations (85.40° E, 19.00° N; 84.95° E, 19.00° N) corresponding to the event.



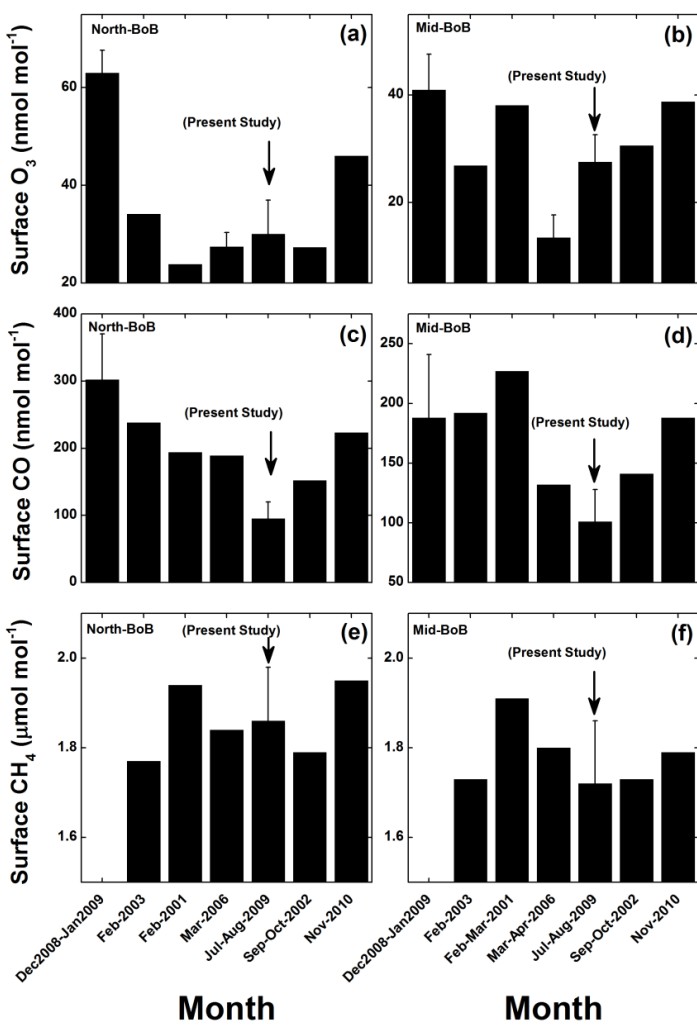


**Figure 16.** Seasonal variation in average $O_3$, CO, and $CH_4$ mixing ratios over (a, c, e) northern BoB and (b, d, f) central BoB (see Fig. 3 for demarcation of these two regions). Except for July–August 2009 (present study period), all average values are obtained from the literature (David et al., 2011; Lal et al., 2007; Lal et al., 2006; Nair et al., 2011; Srivastava et al., 2012; Sahu et al., 2006; and Mallik et al., 2013). Error bars show standard deviations for 825 respective study periods. For any points for which high resolution measurements are not available, standard deviations are not shown.