# Peer review of "Variations in O3, CO, and CH4 over the Bay of Bengal during the summer monsoon season: Ship-borne measurements and model simulations"

_Atmospheric Chemistry and Physics, 2016_

## Referee Comment (RC1) · Anonymous Referee #1 · 10 Aug 2016

Manuscript Reference: Atmos. Chem. Phys. Discuss., doi:10.5194/acp-2016-595, 2016 Title: Variations in O3, CO, and CH4 over the Bay of Bengal during the summer monsoon season: Ship-borne measurements and model simulations by Girach et al. 2016 General remark: This paper is based on the surface (ship-borne) measurements of ozone, CO and methane over the Bay of Bengal (BoB) during summer/monsoon months of year 2009. The main objective of this study is to investigate the spatio-temporal variations of trace gases. The WRF-chem simulations have been compared. Case studies, mostly reduced levels of ozone during rainfall events have been investigated in details. Inferences from surface measurements over the land (India subcontinent) have been used to study the en route transformation (net O3). The data and

analysis is good but the discussion needs somewhat better and concrete interpretation. The paper may be accepted but following aspects need improved or more considerations. (1) Dividing BoB in regions southern, central and north, etc. is not impressive as transport of air mass is highly variable. I agree with categorization based on the trajectories. Characterizing the air masses measured over coastal and open oceans are valuable. (2) Significant data measured during the stationary phase (of ship) has been used. Typically, researchers reject such data. I think data measured during this period should not be used. (3) En route transformation of ozone has been assessed with reference to several station based measurements over the land. Instead of relying on observations, model data should be used/compared to estimate the en-route transformation of ozone. (4) Methane data have been overlooked, or else, can be removed from the draft. (5) There are too many Figures (16), some data plots are repeated. Therefore, improved representations of figs and tables are also required. (5) There is scope of improving English. Excessive use of "WE" "OUR" "FIRST", etc. is not desirable. Following detailed and specific comments should be considered.

Abstract:

Page 1, Line 32-33 "simulations for a low-O3 event on August 10, 2009.............transporting ozone-rich airmasses" This sentence seems contradictory as low -ozone is explained by transport of ozone rich air?

Introduction:

Page 2, Line 64 "The marine regions adjoining South Asia, despite being far from direct anthropogenic activities," It is not really true, marine regions of AS and BoB are surrounded by polluted land of SEA and SA. If authors like to convey that there are no significant emissions (except ships) then it is well understood and do not require to mention "despite being far from...."

Page 3 line 69: "The airmasses exposed to ......" better to rewrite , "exposed " is not an appropriate choice.

Interactive
comment

Page 3 Line 70-72: "In situ measurements over the.....transformation." This is not well written , please re-write.

Page 3, Line 81-83: "Both the export of ......transport of ...synoptic scale dynamics and monsoonal circulations" This is not well written what is the difference (scientifically) between export and transport.

Page 3, Line 85: "highly conducive for the accumulation of trace species". It is not clear , do the outflows from continents stop over BoB? I mean ACCUMULATION is not an appropriate choice?

Page 3, Line 96: "and an unnamed campaign" this does not sound good, write under "other campaigns "etc.

Pag3, Line: 103 ", which influences the oxidation capacity of the atmosphere" This is not required, in other seasons oxidation capacity can also be influenced due to higher levels of VOCs. So it is not unique for this season.

Pag4, Line: 106 "remote sensing of.." here, there is no need to give examples of TES or AURA, it is in general true for any remote sensing technique.

Page4, Line: 114-115 and elsewhere

"spatial and temporal variations in ozone" should be "spatial and temporal variations of ozone"

Page4, Line: 117 "We investigate ...we have ..greater detail." Please re-write "we" has come twice?

Page4, Line: 118-120: This may be deleted

2. The cruise track and background conditions.

Page 4, Line 123-124: Revise as "Figure 1 shows the cruise track of the Oceanic Research Vessel (ORV) Sagar Kanya during the CTCZ campaign (cruise number SK-

261).

This section is partly explained in a rather lengthy caption (Fig1). Try to adjust most explanations in this text sections but not in captions.

"To take time series measurements, the ship was kept stationary at 89° E, 19° N for fifteen days (July 22 to August 06, 2009)."

Usually researchers reject the measurements when ship is stationary , as it is proved that exhaust from ship influences the measurements of trace constituents. An explanation is required how it is ensured that ship's exhaust did not influence measurements around the measurement location?

Line 129-131: "The average prevailing wind patterns at 925 hPa during the cruise period are obtained from NCEP/NCAR reanalysis (http://www.esrl.noaa.gov/psd; Fig. 1). The prevailing westerly or south-westerly winds are conducive for the transport of ozone and its precursors from the Indian landmass to the BoB during the summer monsoon season." revise as here: "The average wind pattern at 925 hPa (NCEP/NCAR reanalysis; http://www.esrl.noaa.gov/psd) during the cruise period is shown in Figure 1. The prevailing westerly and southwesterly winds transport ozone and its precursors from the Indian landmass to the BoB during study period."

Line 133: Phase B (INTEX-B), Does this correspond to Indian summer season?

Line 134: Relatively high NOx emissions are located over parts of eastern and southern India.? This is arguable if compared with emissions over IGP and western India (which is not shown). So change the sentence accordingly.

3. Experimental details and data

Page 5, Line 146 "This instrument was based on the principle.." something like this is better "This instrument works on the principle.." Page 5, Line 135-154 "Trace gas measurements affected by the ship exhaust ....." This is the issue when ship is stationary, irrespective of wind direction.

Page 5, Line 164-165 following should be better "at 5-minute of integration time using an automatic weather..."

So far, it is not clear why measurements at Thumba, Thiruvananthapuram have been discussed in this draft. Objective of using the Thumba data is missing.

5. Results and Discussion

Page 6 Line 205: "period of the summer monsoon season" can be deleted. "summer season" is being repeated again and again.

Page 6 Line 206-207: revise as here

"The mixing ratios of trace gases show large spatio-temporal variations over the BoB."

Page 6 Line 207-209: can be summarized as "Levels of O3 and CO varied in the ranges of 8-54 nmol mol-1 (with average of 29.7±6.8 nmol mol-1) and 50-198 nmol mol-1 ( average of 96±25 nmol mol-1), respectively. "

Page 7, line 220-221: " In addition to sailing across the BoB....." use of stationary ship data is questionable, strong justification is required or remove this data.

Page 7, line 238-242: " Similar variations in O3 mixing ratios and residence time over continental India indicate the influences of transport from ................ and en route photochemistry."

This does not go well. In this paper, it is explained that there is en route formation of ozone, so ozone is formed also over the oceanic region. Therefore this relationship study between residence time (over land) and a secondary species is meaningless. However, to some extent it is meaningful for primary pollutants such as CO and also for CH4. These processes: source strengths, vertical mixing or dilution, and en route photochemistry (or their variability) are not occasional but are continuous. Therefore, and overall, consistent discussion using the residence time calculation is required. It is expected to see best relation between CO and residence time, at least better than O3.

Why CH4 is not influenced by the change in residence time?

Page 7-8, line 250-287: This discussion should be shortened, this lacks completeness. All inferences have been derived using two point measurements at Thumba and Anantpur. I feel, if model is doing good job then rely on model data for such discussion. Otherwise question may arise ; 1. Os the distribution of ozone and southern India homogeneous over central/southern India? 2.Is en-route transport from India the only factor controlling ozone over BOB? downdraft of O3-rich/poor air, or transport from other regions such as SEA are not relevant? Therefore, detailed discussion insightful analysis is required, otherwise, just shorten this part. 5.2 WRF-Chem simulations:

Throughout the draft: It is not nice to see frequent use of "we", "our" and "first time". please try to minimize the use of such words.

Page 9, line 292-293: "variations in the meteorological parameters simulated by the model are briefly evaluated" This analysis is beyond the scope of this paper, I suggest to remove Figure 8.

Page 9, line 312: " that is the mean value subtracted from the mean diurnal pattern,.." This is not clear?

Page 9, line 315: "Ship exhaust contaminated the observations for a period of time between 5 to 14 hours long;.." Here is the entire issue of using stationary data. Questions: How it is ensured that rest of hours were not impacted by ship exhaust? why this period (5-14 hr) is fixed on each day? Second, how about residual air mass (aging of ship exhaust), which can definitely change photochemistry of O3 during rest of the hours. Suggestion: Do not use stationary phase data, you have got great deal of other results to focus on.

Page 10, line345-346: "We suggest that, in the presence of ozone-poor airmass aloft, a downdraft would result in reductions in surface ozone mixing ratios." This is reasonable but if mid-troposheric air (typically O3 higher than at surface) is down-drafted then one

may have opposite scenario. This needs to be mentioned.

Page 12-13, line 432-442: The cause of seasonality is explained in very general terms, and these are well known. Or at least new insights have not been presented about the seasonality .I suggest to remove this part.

Table: Table1: Only references are not enough. Please prove unique/salient features of the option for different atmospheric processes in second column and third can be used for the references. Little more elaborated table is required which will justify the options used for atmospheric processes.

Table 2: Revise as here: "Table 2. A comparison of average surface O3 mixing ratios measured at various sites during summer monsoon period.*boundary layer ozone over the Arabian Sea. " Also Arrange the Table properly, for example, Ahmedabad data are coming at 3 different rows. Better to show "mean +/-1-sigma format" rather showing mean and 1-sigma in two different columns.

Table 4: Last row "No name" , better leave it blank

Figure 1. Caption is very lengthy. What is the unit of wind speed (m/s)? Is NOx data corresponding to period of observations? Revise the map so that NOx emission over entire (southern/central) continents is covered. This is required as back trajectories pass through the beyond the domain shown in present map.

Figure 2. Revise the map as suggested for Figure1 (to show the distributions of trace gases over entire southern and central India).

Figure 4. The color scale should be further resolved (1km is not good enough, at least 500m would be better), as I only see the red (mostly). Also show symbol along the trajectories for each back day. This will help to understand "residence hour" calculation.

Figure 5. Why UT is used in time series plot, while LT is used in a diurnal plot? Better to use LT in all plots (other figures also). Again captions are too long, legends and colors are good enough, no need to mention or repeat same in text (caption).

Figure 6. This figure is not impressive and not required (already you have 16 figs). Instead, A few lines in text should be okay.

Figure 8. This is redundant figure, already results have been summarized in Table 3. I suggest to remove this Figure.

Figures 5 and 9 can be combined: I do not understand why same data (residual hours) have been plotted in 3 different panels. Keep just one (may be in bottom panel). Instead of residual hours (right y-axis), plot WRF chem results.

Figure 10. Again, there is need to shorten the caption, do not explain the legends in details.

Figures 11 and 12 can be merged: Instead of Figure 12, plot temp data in left-y axis in Fig11. No need for Figure 12.

---

## Referee Comment (RC2) · Anonymous Referee #2 · 15 Aug 2016

This manuscript presents measurements of O3, CO and CH4 made from a ship sailing in the Bay of Bengal (BoB) during 2009. The work investigates the spatio-temporal variation of these trace gases, looking at the relationship between their observed mixing ratios and air mass origin and also investigates how well WRF-chem simulations can reproduce the observations. The paper is suitable for ACP and should be accepted subject to the following minor revisions:

General comments: Section 5.1: The first part of the analysis looks at the variation in concentrations of the trace gases along the cruise track and attempts to explain them by looking at air mass origin (the % of residence time over land). This is shown nicely in figure 5, however I feel figures 2 and 3 could be merged (in general the paper has

too many figures). The data seems to be divided into two regions (central and northern BoB) and I am not sure this is necessary. The difference in data taken in different areas in more likely to be driven by air mass origin rather than the area that the ship was in so I would stick to this analysis.

Section 5.2: In general I feel this section could be expanded. Why has CH4 data not been investigated with the model here? From figure 5 it seems that there is reasonable agreement between the observed CH4 and residence time over land of the air so it would have been interesting to see how well the model reproduced the CH4. In general CH4 data is often overlooked in the paper, even though the dataset seems reasonably complete and CH4 is mentioned in the title. If the authors are not confident in the CH4 measurements then they should be removed. Can the authors comment on the main in source of the increased ozone (e.g. anthropogenic / biogenic emissions). What levels of NOx are seen in the model? The comparison of meteorological parameters from the model does not add much to the analysis and the authors should consider removing it (which helps reduce the number of figures). Could the authors also compare model data to the measurements at the surface sites? This would help assess how well the model predicts the air coming into the region and whether this contributes to any discrepancies in the data after emissions and processing.

Section 5.3: It seems that much of the data here has had to be removed due to contamination from the ship exhaust. This causes a large gap in the diurnal average where there is no data between 0600 and 1300, a time of particular interest for photochemistry. Because of this the authors should consider removing this analysis.

Section 5.4: Figures 11 and 12 seem to essentially show the dame thing – could the authors combine them somehow.

Section 5.5: The seasonal variation is investigated by examining data from a series of previous publications of measurements in the region, presented in table 3. The analysis here is good, however I find table 3 hard to interpret. Could the data presented as a

figure?

Minor comments:

Both 'O3' and 'ozone' are used throughout the text. The authors should pick one and stick to it.

Line 151: How were the analyses calibrated? A few lines of detail and references should be given here.

The authors should try to avoid excessive use of the terms 'we' and 'our' when describing the results.

Figure 4 is very hard to interpret – could the authors find a clearer way of showing air mass origin for the different positions on the cruise track?

---

## Author Comment (AC1) · 17 Nov 2016

Please find the response to referee comments and manuscript in the Supplement zip file.

Please also note the supplement to this comment:
http://www.atmos-chem-phys-discuss.net/acp-2016-595/acp-2016-595-AC1-supplement.zip

---

## Author Response (AR1)

**Response to the comments of Reviewer #1**

General remark: This paper is based on the surface (ship-borne) measurements of ozone, CO and methane over the Bay of Bengal (BoB) during summer/monsoon months of year 2009. The main objective of this study is to investigate the spatiotemporal variations of trace gases. The WRF-chem simulations have been compared. Case studies mostly reduced levels of ozone during rainfall events have been investigated in details. Inferences from surface measurements over the land (India subcontinent) have been used to study the en route transformation (net O3). The data and analysis is good but the discussion needs somewhat better and concrete interpretation. The paper may be accepted but following aspects need improved or more considerations.

We thank the reviewer for careful evaluation of the paper. The paper is suitably revised by incorporating the reviewer's suggestions and comments. Please note that the line number mentioned in the reply is corresponding to the revised version ("CTCZ-BOB-R1\_Track\_Changed").

(1) Dividing BoB in regions southern, central and north, etc. is not impressive as transport of air mass is highly variable. I agree with categorization based on the trajectories. Characterizing the air masses measured over coastal and open oceans are valuable.

Following reviewer's suggestion, we removed the discussion based on BoB region's division (Page: 1, Lines: 21-26; Page 7, Line: 244-248) and primarily used categorization based on trajectories. The computations of region wise mean values are only used for comparison with other seasons (subsection 5.4) obtained from previous papers. Such consistency of region is required for comparisons considering strong spatial variability over the BoB, more pronounced during winter (David et al., 2011, Nair et al., 2011)

(2) Significant data measured during the stationary phase (of ship) has been used. Typically, researchers reject such data. I think data measured during this period should not be used.

A contamination of ship-based observation could occur when ship is stationary and winds are calm. Samples taken upwind are not contaminated when winds are strong carrying the exhaust downwind. Here, when ship is stationary and wind speed is above  $5 \text{ ms}^{-1}$  in a direction opposite to the instruments as if ship is in motion relative to the air getting sampled, observations are not affected.

To further discard any effects, we used continuous NOx measurements as tracers of ship exhaust. Though NOx observations are not reported in the paper as the NOx levels were mostly below or around the detection limit of the instrument (1 ppbv), except during the effects of ship exhausts. A case of data filtering is shown below:

Figure shows variation of  $O_3$  and  $NO_x$  on July 31, 2009. It is clearly seen that  $NO_x$  level was ~1 ppbv up to 05:00 hours and abruptly increases to 100 ppb reaching up to 600 ppb under the influence of ship-exhaust, this clearly discriminate the contamination. We have discarded the observations for such contaminations. This exhaust episode occurred when ship was rotated for oceanographic measurements.

(3) En route transformation of ozone has been assessed with reference to several station based measurements over the land. Instead of relying on observations, model data should be used / compared to estimate the en-route transformation of ozone.

We conduct simulations on a larger domain to estimate the en route ozone formation. Model simulations indicate that ozone production rate is about 4.6 nmol mol-1 day-1. (Fig. 7; page-11, line: 381-387)

(4) Methane data have been overlooked, or else, can be removed from the draft.

As replied also to referee #2, we have extended the Methane discussion. Despite of longer chemical lifetime, observed spatial heterogeneity in  $CH_4$  highlights the influence of transport from different source regions located in India to the BoB during the summer monsoon. Further analysis using  $CH_4$  retrievals from SCIAMACHY have been conducted, showing that stronger methane sources are located in central/northern Indian region as compared to southern India (new Figure 6). This is, in agreement with the result based on trajectory assisted analysis, showing that air masses from the central India have significantly higher  $CH_4$  over the BoB than other air masses. The discussion is suitably revised in the manuscript (Figure 6; Page 10, Lines 328-343). It is further inferred from sector-wise analysis of emissions over the hotspot region (i.e. eastern IGP) that these higher  $CH_4$  emissions are due to rice cultivation, waste treatment and livestock (Page: 10, Lines 338-339).

Modeling CH4 from the WRF-Chem setup used here was not possible as this tracer is updated only through chemical boundary conditions from a global model (MOZART). Nevertheless the observational values are presented here as reference for future studies. The correlation between

the presented in situ  $CH_4$  measurements with retrievals from AIRS satellite instrument was found to be statistically insignificant (not shown) which further highlights the need of reporting in situ measurements from this region. This is also discussed in the revised manuscript (Page 10, Lines: 339-343).

(5) There are too many Figures (16), some data plots are repeated.

Therefore, improved representations of figs and tables are also required.

Following reviewer's suggestion, representation of figures and tables has been improved and total number of figures has been reduced in the revised manuscript (from 16 to 12).

(6) There is scope of improving English. Excessive use of "WE" "OUR" "FIRST", etc. is not desirable.

Use of "we", "our" and "first" has been minimized. English improvement has been made.

Following detailed and specific comments should be considered.

**Abstract:**

Page 1, Line 32-33 "simulations for a low-O3 event on August 10, 2009.....transporting ozone-rich airmasses" This sentence seems contradictory as low -ozone is explained by transport of ozone rich air?

Sentence is suitably modified.

**Introduction:**

Page 2, Line 64 "The marine regions adjoining South Asia, despite being far from direct anthropogenic activities," It is not really true, marine regions of AS and BoB are surrounded by polluted land of SEA and SA. If authors like to convey that there are no significant emissions (except ships) then it is well understood and do not require to mention "despite being far from...." As suggested, the segment "despite being far from direct anthropogenic activities," is removed.

Page 3 line 69: "The airmasses exposed to ......" better to rewrite, "exposed" is not an appropriate choice.

Thanks for the suggestion. "Exposed" replaced by "influenced".

Page 3 Line 70-72: "In situ measurements over the.....transformation." This is not well written, please re-write.

Sentence is re-written as "In situ measurements over the marine regions are required to understand the effects of direct outflow, en route chemical transformation, and the chemistry in the transported airmasses".

Page 3, Line 81-83: "Both the export of .....transport of ...synoptic scale dynamics and monsoonal circulations" This is not well written what is the difference (scientifically) between export and transport.

The sentence is re-written as "Transport of airmasses between Indian subcontinent and adjacent marine regions has strong seasonal dependence associated with the monsoonal circulation"

Page 3, Line 85: "highly conducive for the accumulation of trace species". It is not clear, do the outflows from continents stop over BoB? I mean ACCUMULATION is not an appropriate choice?

"accumulation" word is removed and sentence is suitably modified.

Page 3, Line 96: "and an unnamed campaign" this does not sound good, write under "other campaigns "etc. Suggestion is incorporated.

Pag3, Line: 103 ", which influences the oxidation capacity of the atmosphere" This is not required, in other seasons oxidation capacity can also be influenced due to higher levels of VOCs. So it is not unique for this season. Yes, modified accordingly

Pag4, Line: 106 "remote sensing of.." here, there is no need to give examples of TES or AURA, it is in general true for any remote sensing technique. Suggestion is incorporated.

Page4, Line: 114-115 and elsewhere "spatial and temporal variations in ozone" should be "spatial and temporal variations of ozone" Suggestion is incorporated.

Page4, Line: 117 "We investigate ...we have ..greater detail." Please re-write "we" has come twice?

Suggestion is incorporated.

Page4, Line: 118-120: This may be deleted Suggestion is incorporated.

**2. The cruise track and background conditions.**

Page 4, Line 123-124: Revise as "Figure 1 shows the cruise track of the Oceanic Research Vessel (ORV) Sagar Kanya during the CTCZ campaign (cruise number SKC3 261).

Sentence is revised as suggested, except "SKC3 261". The cruise number "SK 261" given by NCAOR (National Centre for Antarctic and Ocean Research), Goa, India is used.

This section is partly explained in a rather lengthy caption (Fig1). Try to adjust most explanations in this text sections but not in captions.

The caption of Figure 1 is shortened significantly and explanation is given now in the section 2.

"To take time series measurements, the ship was kept stationary at 89\_ E, 19\_ N for fifteen days (July 22 to August 06, 2009)." Usually researchers reject the measurements when ship is stationary, as it is proved that exhaust from ship influences the measurements of trace constituents. An explanation is required how it is ensured that ship's exhaust did not influence measurements around the measurement location?

Please see the response corresponding major comment #2.

Line 129-131: "The average prevailing wind patterns at 925 hPa during the cruise period are obtained from NCEP/NCAR reanalysis (http://www.esrl.noaa.gov/psd; Fig. 1). The prevailing westerly or south-westerly winds are conducive for the transport of ozone and its precursors from the Indian landmass to the BoB during the summer monsoon season." revise as here: "The average wind pattern at 925 hPa (NCEP/NCAR reanalysis; http://www.esrl.noaa.gov/psd) during the cruise period is shown in Figure 1. The prevailing westerly and southwesterly winds transport ozone and its precursors from the Indian landmass to the BoB during the Indian landmass to the BoB during the summer monsoon season." revise as here: "The average wind pattern at 925 hPa (NCEP/NCAR reanalysis; http://www.esrl.noaa.gov/psd) during the cruise period is shown in Figure 1. The prevailing westerly and southwesterly winds transport ozone and its precursors from the Indian landmass to the BoB during study period." Suggestion is incorporated.

Line 133: Phase B (INTEX-B), Does this correspond to Indian summer season?

INTEX-B inventory provides annual mean emissions. References using INTEX-B inventory are provided in the manuscript (Page 6-7, Lines 214-216).

Line 134: Relatively high NOx emissions are located over parts of eastern and southern India.? This is arguable if compared with emissions over IGP and western India (which is not shown). So change the sentence accordingly.

Sentence modified accordingly, NOx emissions are higher over eastern and southern India as compared to that of central Indian.

**3.** Experimental details and data**

Page 5, Line 146 "This instrument was based on the principle.." something like this is better "This instrument works on the principle.."

Corrected accordingly.

Page 5, Line 135-154 "Trace gas measurements affected by the ship exhaust ....." This is the issue when ship is stationary, irrespective of wind direction.

In strong winds, measurements made upwind direction are not affected by ship exhaust, Nevertheless additional filtering have been implemented as mentioned in the response to major comment #2.

Page 5, Line 164-165 following should be better "at 5-minute of integration time using an automatic weather..."

Sentence modified accordingly.

So far, it is not clear why measurements at Thumba, Thiruvananthapuram have been discussed in this draft. Objective of using the Thumba data is missing. The suggested discussion is added in revised manuscript (Page-6, Lines 183-184).

**5. Results and Discussion**

Page 6 Line 205: "period of the summer monsoon season" can be deleted. "summer season" is being repeated again and again. It is deleted.

Page 6 Line 206-207: revise as here "The mixing ratios of trace gases show large spatio-temporal variations over the BoB." Revised accordingly. Page 6 Line 207-209: can be summarized as "Levels of O3 and CO varied in the ranges of 8-54 nmol mol-1 (with average of 29.7±6.8 nmol mol-1) and 50-198 nmolmol-1 (average of 96±25 nmol mol-1), respectively. " Changed accordingly.

Page 7, line 220-221: " In addition to sailing across the BoB....." use of stationary ship data is questionable, strong justification is required or remove this data. Please see the response corresponding major comment #2.

Page 7, line 238-242: "Similar variations in O3 mixing ratios and residence time over continental India indicate the influences of transport from ...... and en route photochemistry." This does not go well. In this paper, it is explained that there is en route formation of ozone, so ozone is formed also over the oceanic region. Therefore this relationship study between residence time (over land) and a secondary species is meaningless. However, to some extent it is meaningful for primary pollutants such as CO and also for CH4. These processes: source strengths, vertical mixing or dilution, and en route photochemistry (or their variability) are not occasional but are continuous. Therefore, and overall, consistent discussion using the residence time calculation is required. It is expected to see best relation between CO and residence time, at least better than O3.

We have revised the calculation of residence time by taking only those hours when air parcels is typically within an altitude of 1.5 km over land, as residence of air masses over continent (but aloft) might not get direct influences of surface emissions. The revised analysis shows better correlation for primary species (R = 0.4 for both CO and CH4) and slightly lower for secondary species (R=0.3 for O3). The manuscript has been suitably revised to incorporate these changes (new Fig. 4 and Page 8, line 272-277).

Why CH4 is not influenced by the change in residence time?

Please also see the response to your previous comment.  $CH_4$  is also found to be influenced by residence time as shown in the revised analysis (Fig-4, page-8, line 276-277).

Page 7-8, line 250-287: This discussion should be shortened, this lacks completeness. All inferences have been derived using two point measurements at Thumba and Anantpur. I feel, if model is doing good job then rely on model data for such discussion. Otherwise question may arise ; 1. Os the distribution of ozone and southern India homogeneous over central/southern India? 2. Is en-route transport from India the only factor controlling ozone over BOB? downdraft of O3-rich/poor air, or transport from other regions such as SEA are not relevant? Therefore, detailed discussion insightful analysis is required, otherwise, just shorten this part.

We agree that estimation here has been based on very limited observations. Following reviewer's suggestion, now we investigated the en route ozone production by analyzing model simulated  $O_3$  along the air mass trajectories at several representative locations in the BoB.

Based on modeled  $O_3$  along airmass trajectories ending over BoB, we find that ozone production rate is about 4.6 nmol mol-1. The new analyses and discussions are incorporated in revised manuscript (new Fig. 7; Page:11, Lines: 380-387)

**5.2 WRF-Chem simulations:**

Throughout the draft: It is not nice to see frequent use of "we", "our" and "first time". please try to minimize the use of such words.

Thanks for the suggestion. The use of "we", "our" and "first" is minimized.

Page 9, line 292-293: "variations in the meteorological parameters simulated by the model are briefly evaluated" This analysis is beyond the scope of this paper, I suggest to remove Figure 8. Figure is removed as suggested. Comparison between simulated and observed meteorological parameters is mentioned in one sentence.

Page 9, line 312: " that is the mean value subtracted from the mean diurnal pattern,..." This is not clear?

Following reviewer#2's suggestion, this section (Diurnal variation) is removed in the revised manuscript.

Page 9, line 315: "Ship exhaust contaminated the observations for a period of time between 5 to 14 hours long;.." Here is the entire issue of using stationary data. Questions:

How it is ensured that rest of hours were not impacted by ship exhaust? Why this period (5-14 hr) is fixed on each day? Second, how about residual air mass (aging of ship exhaust), which can definitely change photochemistry of O3 during rest of the hours. Suggestion: Do not use stationary phase data, you have got great deal of other results to focus on. How it is ensured that rest of hours were not impacted by ship exhaust?

The ship was rotated for oceanographic measurements, typically conducted between 5–14 hours making sampling inlet downwind ship exhaust. Please note that the section being referred to showing the average diurnal variation has been removed following reviewer#2's suggestion

Second, how about residual air mass (aging of ship exhaust), which can definitely change photochemistry of O3 during rest of the hours.

The direct influences of exhaust are swept away by strong southwesterly winds. As wind direction remained same, there would be minimal effects of the aging downwind on the upwind measurements, residual effects being similar to what would be caused by mixing of the background with emissions caused by ship transportation in the BoB.

Page 10, line345-346: "We suggest that, in the presence of ozone-poor airmass aloft, a downdraft would result in reductions in surface ozone mixing ratios." This is reasonable but if mid-troposheric air (typically O3 higher than at surface) is down-drafted then one may have opposite scenario. This needs to be mentioned.

This is mentioned now. (Page-12, lines 424-425)

Page 12-13, line 432-442: The cause of seasonality is explained in very general terms, and these are well known. Or at least new insights have not been presented about the seasonality. I suggest removing this part.

Here, we wanted to show how the measurements during CTCZ experiment fills the gap in seasonal variation of trace gases over BoB. This is especially important considering that it is minima of CO seasonal cycle during the monsoon so the CTCZ measurements complete the

information to get overall seasonal variability. Also following the comment of Reviewer#2, we are retaining this section.

**Table:**

Table1: Only references are not enough. Please prove unique/salient features of the option for different atmospheric processes in second column and third can be used for the references. Little more elaborated table is required which will justify the options used for atmospheric processes. Table is revised and an extra column about the features of the schemes is added.

Table 2: Revise as here: "Table 2. A comparison of average surface O3 mixing ratios measured at various sites during summer monsoon period.\*boundary layer ozone over the Arabian Sea. " Also Arrange the Table properly, for example, Ahmedabad data are coming at 3 different rows. Better to show "mean +/-1-sigma format" rather showing mean and 1-sigma in two different columns.

Table 2 and its caption are revised accordingly.

Table 4: Last row "No name", better leave it blank Suggestion is incorporated.

**Figure**

Figure 1. Caption is very lengthy. What is the unit of wind speed (m/s)? Is NOx data corresponding to period of observations? Revise the map so that NOx emission over entire (southern/central) continents is covered. This is required as back trajectories pass through the beyond the domain shown in present map.

Figure 1 is revised accordingly. Caption is shortened and detail is mentioned in the text. NOx emissions are shown over India covering southern and central regions.

Figure 2. Revise the map as suggested for Figure1 (to show the distributions of trace gases over entire southern and central India).

Figure 2 is revised accordingly. Distributions of trace gases over entire southern and central India are shown.

Figure 4. The color scale should be further resolved (1km is not good enough, at least 500m would be better), as I only see the red (mostly). Also show symbol along the trajectories for each back day. This will help to understand "residence hour" calculation.

Figure is revised accordingly. Colour scale is resolved to 500m. The cross symbol along the trajectories for each day is also shown. (Please see Fig. 3)

Figure 5. Why UT is used in time series plot, while LT is used in a diurnal plot? Better to use LT in all plots (other figures also). Again captions are too long, legends and colors are good enough, no need to mention or repeat same in text (caption).

LT is used now in all the plots as suggested. Caption has been shortened.

Figure 6. This figure is not impressive and not required (already you have 16 figs). Instead, A few lines in text should be okay. This figure is removed.

Figure 8. This is redundant figure, already results have been summarized in Table 3. I suggest to remove this Figure. This figure is removed.

Figures 5 and 9 can be combined: I do not understand why same data (residual hours) have been plotted in 3 different panels. Keep just one (may be in bottom panel). Instead of residual hours (right y-axis), plot WRF chem results.

Figure 5 and 9 are combined now. While residence time is shown in bottom panel, WRF chem results are shown corresponding right y-axis. (See Figure 4)

Figure 10. Again, there is need to shorten the caption, do not explain the legends in details. This caption is also shortened.

Figures 11 and 12 can be merged: Instead of Figure 12, plot temp data in left-y axis in Fig11. No need for Figure 12.

Figure 11 and 12 are merged. (See Figure 8)

**Response to the comments of Reviewer #2**

This manuscript presents measurements of  $O_3$ , CO and CH4 made from a ship sailing in the Bay of Bengal (BoB) during 2009. The work investigates the spatio-temporal variation of these trace gases, looking at the relationship between their observed mixing ratios and air mass origin and also investigates how well WRF-chem simulations can reproduce the observations. The paper is suitable for ACP and should be accepted subject to the following minor revisions:

We thank the reviewer for careful evaluation of the manuscript and valuable comments. All the comments and suggestions are incorporated as discussed below. Please note that the line number mentioned in the reply is corresponding to the revised manuscript "CTCZ-BOB-R1\_Track\_Changed".

**General comments:**

**Section 5.1:** The first part of the analysis looks at the variation in concentrations of the trace gases along the cruise track and attempts to explain them by looking at air mass origin (the % of residence time over land). This is shown nicely in figure 5, however I feel figures 2 and 3 could be merged (in general the paper has too many figures).

Figure 2 and 3 are merged in revised manuscript (please see Figure 2). Number of figures is now reduced to 12 (from 16).

The data seems to be divided into two regions (central and northern BoB) and I am not sure this is necessary. The difference in data taken in different areas in more likely to be driven by air mass origin rather than the area that the ship was in so I would stick to this analysis.

We have removed the discussion based on BoB region's division (Page: 1, Lines: 21-26; Page 7, Line: 244-248) and primarily used categorization based on trajectories (see also comments to reviewer#1). However, the computations of region wise mean values are only used for comparison with other seasons (subsection 5.4) to be consistent with previous papers. Such consistency is required for comparisons considering strong spatial variability over the BoB more pronounced during winter (David et al., 2011, Nair et al., 2011)

Section 5.2: In general I feel this section could be expanded. Why has  $CH_4$  data not been investigated with the model here? From figure 5 it seems that there is reasonable agreement between the observed  $CH_4$  and residence time over land of the air so it would have been interesting to see how well the model reproduced the  $CH_4$ . In general  $CH_4$  data is often overlooked in the paper, even though the dataset seems reasonably complete and  $CH_4$  is mentioned in the title. If the authors are not confident in the  $CH_4$  measurements then they should be removed.

The section is expanded by adding the analysis of the influence of India's anthropogenic emissions to  $O_3$  over BoB using model sensitivity simulation (See revised Fig 4, Page-11, Lines: 375-379). CH4 from WRF-Chem was not analyzed as the existing model setup did not include explicit treatment of regional emissions of CH4, being included through chemical boundary conditions from global model MOZART. Nevertheless the observational values are presented here for their use in future studies. To investigate the spatial variability in observed CH4, retrievals of SCIAMACHY are now analyzed (Fig. 6) which reveals higher methane concentration over central Indian region compared to southern Indian region during the study period, complementing the trajectory assisted analysis.

We are confident that our measurements and  $CH_4$  data are reasonably complete to derive the spatial variation. Interestingly, despite of longer chemical lifetime, the observed spatial heterogeneity in  $CH_4$  highlights the importance of transport from different source regions located in India to the BoB during the summer monsoon. It is further inferred from sector-wise analysis of emissions over the hotspot region (i.e. eastern IGP) that high  $CH_4$  emissions are due to rice cultivation, waste treatment and livestock (Page: 10, Lines 338-339). The relevant discussion is also suitably revised in the manuscript (Page 10, Lines 328-343).

The correlation between presented in situ CH4 measurements with retrievals from AIRS satellite instrument was found to be statistically insignificant (not shown) which further highlights a need of reporting in situ measurements from this region (Page 10, Lines: 339-343).

Can the authors comment on the main in source of the increased ozone (e.g. anthropogenic / biogenic emissions).

We performed additional model simulation by switching off anthropogenic emissions in the model domain. As shown in revised Fig. 4, the spatio-temporal variations in  $O_3$  over the BoB are mainly controlled by the regional anthropogenic emissions over the South Asia. On average,  $O_3$  mixing ratios over the BoB are predicted to be reduced of about 14 nmol mol-1 in lack of anthropogenic emissions in South Asia. The manuscript is suitably revised to include the new analyses and related discussion (Fig. 4, Page-11, Lines 375-379).

What levels of NOx are seen in the model? Mean NOx mixing ratios along the ship are calculated to be  $135 \pm 90$  pmol mol-1.

The comparison of meteorological parameters from the model does not add much to the analysis and the authors should consider removing it (which helps reduce the number of figures). As suggested by the reviewer, we removed the comparison in the revised version of the manuscript.

Could the authors also compare model data to the measurements at the surface sites? This would help assess how well the model predicts the air coming into the region and whether this contributes to any discrepancies in the data after emissions and processing.

Detailed evaluation of WRF-Chem simulated ozone over surface sites in India has been conducted by Kumar et al., (GMD, 2012b). A comparison of  $O_3$  measurements at Thumba with model showed a good agreement ( $R^2$ =0.6) and mean values compared typically within 1-standard deviation. We have shown surface  $O_3$  is simulated within the 1-sigma variation at another stations (Gadanki) in southern India during monsoon (Ojha et al., 2016, Fig-8). This information is now provided in the revised manuscript (Page: 11; lines: 369-374).

Section 5.3: It seems that much of the data here has had to be removed due to contamination from the ship exhaust. This causes a large gap in the diurnal average where there is no data between 0600 and 1300, a time of particular interest for photochemistry. Because of this the authors should consider removing this analysis.

Following reviewer's suggestion, this section is removed in the revised manuscript.

Section 5.4: Figures 11 and 12 seem to essentially show the same thing - could the authors combine them somehow.

Suggestion is incorporated (Please see Fig. 8).

Section 5.5: The seasonal variation is investigated by examining data from a series of previous publications of measurements in the region, presented in table 3. The analysis here is good, however I find table 3 hard to interpret. Could the data presented as a figure?

The data of table 4 (seasonal variation) is already shown in Fig 16 (which is now Fig. 12 in the revised manuscript)

Minor comments:

Both 'O3' and 'ozone' are used throughout the text. The authors should pick one and stick to it. "O3" is now used throughout the revised manuscript.

Line 151: How were the analysers calibrated? A few lines of detail and references should be given here.

Suggestion is incorporated (Page-5, Lines 168-171 and 181).

The authors should try to avoid excessive use of the terms 'we' and 'our' when describing the results.

Excess use of "we" and "our" is avoided in revised manuscript.

Figure 4 is very hard to interpret – could the authors find a clearer way of showing air mass origin for the different positions on the cruise track?

To make the figure clearer now only representative trajectories (instead of all) are shown. Following suggestion of Reviewer #1, a symbol is added along to trajectories representing a time difference of one day. (Please see Fig. 3)

**Variations in O3, CO, and CH4 over the Bay of Bengal during the summer monsoon season: Ship-borne measurements and model simulations**

5 Imran A. Girach1,2, Narendra Ojha2, Prabha R. Nair1, Andrea Pozzer2, Yogesh K. Tiwari3, K. Ravi Kumar4,5, and Jos Lelieveld2

1Space Physics Laboratory, Vikram Sarabhai Space Centre, Thiruvananthapuram 695022, India

2Department of Atmospheric Chemistry, Max Planck Institute for Chemistry, Mainz 55128, Germany

3Indian Institute of Tropical Meteorology, Pune 411 008, India

10 4National Institute of Polar Research, Tachikawa, Japan

5Department of Environmental Geochemical Cycle Research, JAMSTEC, Yokohama, Japan

Correspondence to: Imran A. Girach (imran.girach@gmail.com) and Narendra Ojha (narendra.ojha@mpic.de)

**15 Abstract**

We present ship-borne measurements of surface ozone  $(O_3)$ , carbon monoxide (CO) and methane  $(CH_4)$  over the Bay of Bengal (BoB), the first time such measurements have been performedtaken during the summer monsoon season, as a part of the Continental Tropical Convergence Zone (CTCZ) experiment during 2009. O3, CO, and CH4 mixing ratios exhibited significant spatial and temporal variability in the ranges of 8–54 nmol mol-1, 50–200 nmol

- 20 mol-1, and 1.57–2.15 μmol mol-1, with means of 29.7±6.8 nmol mol-1, 96±25 nmol mol-1, and 1.83±0.14 μmol mol-1, respectively. While the airmasses were mainly from northern or central India over northern BoB, they were from southern India over central region of BoB. The average mixing ratios of trace gases over northern-BoB in airmasses from northern or central/northern India (O3: 30±7 nmol mol-1, CO: 95±25 nmol mol-1, CH4: 1.86±0.12 μmol mol-1), in airmasses from northern or central India, did not differ much were not statistically different from those in
- 25 airmasses from southern Indiaover central BoB (O3: 27±5 nmol mol-1, CO: 101±27 nmol mol-1, CH4: 1.72±0.14  $\mu$ mol mol-1), in airmasses from southern India. Spatial variability is observed to be most significant for CH4 with higher mixing ratios in the airmasses from central/northern India, where higher CH4 levels are seen in the SCIAMACHY (SCanning Imaging Absorption spectroMeter for Atmospheric CartograpHY) data. n mean O3rate of about 4.6 nmol mol-1 day-1 The ship-based observations, in conjunction with backward air trajectories and ground-
- 30 based measurements over the Indian region, are analyzed to estimate a net  $ozone \underline{O}_3$  production of 1.5–4 nmol mol-1 day-1-in the outflow. Ozone  $\underline{O}_3$  mixing ratios over the BoB showed large reductions (by ~20 nmol mol-1) during four rainfall events. Temporal changes in the meteorological parameters, in conjunction with ozone  $\underline{O}_3$  vertical profiles, indicate that these low ozone  $\underline{O}_3$  events are associated with downdrafts of free-tropospheric ozone  $\underline{O}_3$ -poor airmasses.

While the observed variations inof  $O_3$  and CO are successfully reproduced using the Weather Research and Forecasting model with Chemistry (WRF-Chem), this model overestimates mean concentrations by about 6 and 1620% for  $O_3$  and CO respectively, generally overestimating  $O_3$  mixing ratios during the rainfall events. An analysis of modeled  $O_3$  along airmass trajectories show mean en route  $O_3$  production rate of about 4.6 nmol mol-1 day-1 in the outflow towards the BoB. Analysis of the chemical various tendencies from model simulations during for a low-  $O_3$  an event on August 10, 2009, captured successfully reproduced by the model, shows the key role of horizontal

40

advection in-rapidly transporting  $\underline{ozoneO_3}$ -rich airmasses from near the coast across the BoB.  $\underline{Our-This}$  study fills a gap in the availability of trace gas measurements over the BoB, and when combined with data from previous campaigns, reveals large seasonal amplitude (~39 and ~207 nmol mol-1 for O3 and CO, respectively) over the northern BoB.

**45 1. Introduction**

[revised manuscript text omitted]

- These first monsoonal observations of O3, CO, and CH4 show significant-large spatio-temporal variability over the BoB, with mixing ratios varying in the range of 8–54 (mean: 29.7±6.8) nmol mol-1, 50–200 (mean: 96±25) nmol mol-1, and 1.57–2.15 (mean: 1.83±0.14) µmol mol-1, respectively. The O37 and CO7 and CH4-mixing ratios in airmasses from central/northern India are slightly higher or comparable (O3: 30±7 nmol mol-1, CO: 95±25 nmol mol-1, CH4: 1.86±0.12 µmol mol-4) over northern BoB as compared to those in airmasses from southern India over central BoB-(O3: 27±5 nmol mol-1, CO: 101±27 nmol mol-1, CH4: 1.72±0.14 µmol mol-4). The difference (-0.14 µmol mol-4) between CH4 mixing ratios in airmasses from southern (1.86±0.12 µ
[revised manuscript text omitted]

| Observation  | Longitude | Latitude | Observation period               | Mean-Surface     | Reference                 |  |  |  |  |
|--------------|-----------|----------|----------------------------------|------------------|---------------------------|--|--|--|--|
| site         | (° E)     | (° N)    | during monsoon                   | Daytime          |                           |  |  |  |  |
|              |           |          | season                           | Ozone O 3 |                           |  |  |  |  |
|              |           |          |                                  | (Mean ±   |                           |  |  |  |  |
|              |           |          |                                  | Standard  |                           |  |  |  |  |
|              |           |          |                                  | Deviation)       |                           |  |  |  |  |
| Arabian Sea  |           |          |                                  |                  |                           |  |  |  |  |
| Arabian Sea  | 69 - 76   | 9_19     | July_August 2002                 | 9                | Alietal 2009              |  |  |  |  |
| Aldolali Sea | 0) = 10   | ) -1)    | July-August 2002                 | ,                | An et al., 2007           |  |  |  |  |
| Ahmedabad    | 72.6      | 23       | July August 2003                 | 25*              | Srivastava et al., 2012   |  |  |  |  |
|              |           |          | <del>2007</del>                  |                  |                           |  |  |  |  |
|              |           |          | Western coast of India           |                  |                           |  |  |  |  |
| Thiruyananth | 76.9      | 85       | August 2009                      | 23+7             | Present Study             |  |  |  |  |
| apuram       | 10.7      | 0.5      | August 2007                      | 23 -1     | Tresent Study             |  |  |  |  |
| apuram       |           |          |                                  |                  |                           |  |  |  |  |
| Thiruvananth | 76.9      | 8.5      | June–August 2008                 | 19 ±6     | David and Nair, 2011      |  |  |  |  |
| apuram       |           |          |                                  |                  |                           |  |  |  |  |
| 17           | 75.4      | 11.0     | L L 2010 2011                    | 11.4             |                           |  |  |  |  |
| Kannur       | /5.4      | 11.9     | July 2010–2011                   | 11 +4     | Nishanth et al., 2014     |  |  |  |  |
| MtAbu        | 72.7      | 24.6     | August 1993–2000                 | 25 +9     | Naja et al., 2003         |  |  |  |  |
| (1.6km amsl) |           |          |                                  |                  |                           |  |  |  |  |
| Ahmedabad    | 72.6      | 23       | July 1991–1995,           | 22 +8,    | Lal et al., 2002 ; |  |  |  |  |
|              |           |          |                                  |                  | Srivastava et al., 2012   |  |  |  |  |
|              |           |          | August 1991–1995 , | 17±4,     |                           |  |  |  |  |
|              |           |          | July–August 2003–                | 25*              |                           |  |  |  |  |
|              |           |          | 2007                             |                  |                           |  |  |  |  |
|              |           |          |                                  |                  |                           |  |  |  |  |
| Ahmedabad    | 72.6      | 23       | August 1991–1995                 | 17               | Lal et al., 2002          |  |  |  |  |
|              | I         | I | Central India                    |           | 1                         |  |  |  |  |
|              |           |          |                                  |                  |                           |  |  |  |  |
| Anantpur     | 77.65     | 14.62    | July 2009                        | 30 +2     | Reddy et al., 2011        |  |  |  |  |

| Eastern coast of India |           |         |                           |              |                        |  |  |
|------------------------|-----------|---------|---------------------------|--------------|------------------------|--|--|
| Bhubaneswar            | 86.4      | 20.5    | June–August 2011–
2012 | 29 ±6 | Mahapatra et al., 2014 |  |  |
| Bay of Bengal          |           |         |                           |              |                        |  |  |
| Bay of
Bengal       | 80.3–90.1 | 11–21.1 | July–August 2009          | 30 ±7 | Present Study          |  |  |

**Table 3.** A comparison of mean values from observations with model-simulated parameters along with the mean bias. The squared correlation coefficients correspond to the linear regression analysis between daily averaged in situ and simulated parameters.

| Parameter                     | Observation | Model (WRF-Chem)                               | Mean bias                  | $\mathbf{R}^2$  |  |
|-------------------------------|-------------|------------------------------------------------|----------------------------|-----------------|--|
|                               |             |                                                |                            |                 |  |
| Pressure (hPa)                | 1001.3±2.1  | 999.0 4 ±2.4 2                   | - <del>2.31.9</del> | 0.93            |  |
| Temperature (°C)              | 29.3±0.9    | 28.8 7 ±0.6                             | -0.5 6              | 0.1 3 2  |  |
| Relative Humidity (%)         | 87.9±4.2    | 86.5 8 ±2.8                             | -1. 1 4             | 0. 36 54 |  |
| $O_3 \text{ (nmol mol}^{-1})$ | 29.7±6.8    | 3 <del>5.91.6±86.36</del> | 6.2 1.9      | 0. 58 67 |  |
| CO (nmol mol -1 )  | 96±25       | 11 48 ±3 0 7                     | 2218                | 0.19            |  |

**Table 4**. A comparison of average mixing ratios of surface trace gases measured over northern BoB (81-91° E, 16-21.5° N) and central BoB (80-91° E, 11-16° N) in different seasons as measured during different experiments. The range of mixing ratios (i.e. minima–maxima) is given in the brackets. \*CO mixing ratios below the detection limit (i.e. 50 nmol mol-1) are not considered in the analysis.

| Study
period | Name of
Experiment | Reference    | O 3 (nmol
mol -1 ) over | O 3 (nmol
mol -1 ) over | CO (nmol
mol -1 ) over | CO (nmol
mol -1 ) | CH 4 (µmol
mol -1 ) | CH 4 (µmol
mol -1 ) |
|-----------------|-----------------------|--------------|--------------------------------------------------|--------------------------------------------------|--------------------------------------|---------------------------------|----------------------------------------------|----------------------------------------------|
|                 |                       |              | northern                                         | central                                          | northern                             | over                            | over                                         | over                                         |
|                 |                       |              | ВОВ                                              | вов                                              | ROR                                  | central
BoB                  | northern
BoB                              | central
BoB                               |
|                 |                       |              |                                                  |                                                  |                                      |                                 |                                              |                                              |
| December        | W ICARB               | David et     | 63 0+4 7                                         | 40 9+6 7                                         | 302+68                               | 188+53                          | No data                                      | No data                                      |
| 2008-           | W_ICIND               | al 2011      | (50 8–73 8)                                      | (27.7-63.5)                                      | (140-450)                            | (50-320)                        | 110 dulu                                     | 110 data                                     |
| Ianuary         |                       | ul., 2011    | (30.0 73.0)                                      | (21.1 05.5)                                      | (110 150)                            | (30 320)                        |                                              |                                              |
| 2009            |                       |              |                                                  |                                                  |                                      |                                 |                                              |                                              |
| February        | BOBEX-II              | Lal et al    | ~34.1                                            | ~26.8                                            | ~238                                 | ~192                            | ~1.77                                        | ~1.73                                        |
| 2003            | 2022.11               | 2007         | (15.8–50.4)                                      | (13.9–35.0)                                      | (187–292)                            | (159–224)                       | (1.70–1.85)                                  | (1.68–1.77)                                  |
| Februarv–       | BOBEX-I               | Lal et al    | ~23.8                                            | ~38.0                                            | ~194                                 | ~227                            | ~1.94                                        | ~1.91                                        |
| March 2001      |                       | 2006         | (16.1–38.3)                                      | (19.4–62.9)                                      | (165–235)                            | (97–339)                        | (1.89–2.02)                                  | (1.74–2.06)                                  |
| March-          | ICARB                 | Nair et al., | 27.4±2.9                                         | 13.4±4.2                                         | ~189                                 | ~132                            | ~1.84                                        | ~1.80                                        |
| April 2006      |                       | 2011;        | (21.4–32.6)                                      | (3.1–24.6)                                       | (157–235)                            | (96–167)                        | (1.80–1.88)                                  | (1.75–1.84)                                  |
|                 |                       | Srivastava   |                                                  |                                                  |                                      |                                 |                                              |                                              |
|                 |                       | et al., 2012 |                                                  |                                                  |                                      |                                 |                                              |                                              |
| July-August     | CTCZ                  | Present      | 30.0±6.9                                         | 27.5±5.0                                         | 95±25 *                              | 101±27 *                        | 1.86 ±0.12                                   | 1.72±0.14                                    |
| 2009            |                       | Study        | (8.50–54.1)                                      | (8.8–40.5)                                       | (50-198)*                            | (50-157)*                       | (1.62–2.15)                                  | (1.57–1.96)                                  |
| September-